# A novel lidar gradient cluster analysis method of nocturnal boundary layer detection during air pollution episodes

**Yingchao Zhang[1], Su Chen[1], Siying Chen[1], He Chen[1], Pan Guo[1]**

[1] School of Optics and Photonics, Beijing Institute of Technology, Beijing 100081, China

*Correspondence to: Siying Chen (csy@bit.edu.cn)*

**Abstract.** The observation of the nocturnal boundary layer height (NBLH) plays an important role in air pollution and monitoring. Through 39-d of heavy pollution observation experiments in Beijing (China), as well as the exhaustive evaluation of the gradient, wavelet covariance transform, and cubic root gradient methods, a novel algorithm based on the cluster analysis of the gradient method (CA-GM) of lidar signals is developed to capture the multilayer structure and achieve

night-time stability. The CA-GM highlights its performance compared with radiosonde data, and the best correlation (0.85), weakest root mean square error (203 m), and an improved 25% correlation coefficient are achieved via the GM. Compared with the 39-d experiments with other algorithms, reasonable parameter selection can help in distinguishing between layers with different properties, such as the cloud layer, elevated aerosol layers, and random noise. Consequently, the CA-GM can automatically address the uncertainty with multiple structures and obtain a stable NBLH with a high temporal resolution,

which is expected to contribute to air pollution monitoring and climatology, as well as model verification.

## 1 Introduction

Air pollution has an important impact on human health, climatic patterns, and the ecological environment (Shi et al., 2019; Su et al., 2020a; Wang et al., 2020). The primary anthropogenic emission source is particulate matter (PM), which is the major source of severe haze in Beijing (Lv et al., 2020; Ma et al., 2019). Many passive and active remote sensing

instruments have been combined to observe aerosol optical and microphysical properties (Ji et al., 2018b; Wang et al., 2019), the relationships between PM and meteorology (Li et al., 2019; Zhang et al., 2015), and aerosol–atmospheric boundary layer height (ABLH) interactions (Dong et al., 2017; Su et al., 2020b). With the development in star and moon photometry, continuous day-to-night detection has improved the estimation of column-integrated aerosol properties at night. (Benavent-Oltra et al., 2019; Pérez-Ramírez et al., 2008). Nevertheless, there are a few experiments are observed in the nocturnal

boundary layer (NBL). The complexity of weak wind forces, significant stratification, and intermittent turbulence (Stull, 1988; Weil, 2011) results in the continuous accumulation of fine particles near the surface. The turbulent mixing process is accompanied with a strong physiochemical effect, which favours the formation of new particles and worse the pollution (Hao et al., 2018; Wang et al., 2018). Therefore, accurately acquiring the nocturnal boundary layer height (NBLH) during a polluted episode, especially at night, is of great significance toward combatting air pollution.

Multiple approaches have been developed to determine the ABLH based on various observations, including radiosounding, remote sensing, and parameterisation from laboratory experiments (Li et al., 2017b; McGrath-Spangler and Denning, 2012; Nakoudi et al., 2019; Su et al., 2020a). The lidar uses an aerosol as a tracer for mixing processes with high space and temporal resolutions (Kumar, 2006; Leventidou et al., 2013; Yuval et al., 2020). The stable condition shows further agreement between lidar and radiosonde than the unstable condition because of the complex aerosol structure that

complicates NBLH retrieval (Emeis and Schäfer, 2006; Martucci et al., 2007; Sawyer and Li, 2013). At night, the NBLH determined by elastic lidar is either the top of the residual layer or the top of the surface mechanically driven mixing layer (Dang et al., 2019a; Yuval et al., 2020). In the absence of external forces, aerosols in the atmosphere become stratified, resulting in single or multiple layers (elevated or advent aerosols) depending on the location and type of the atmospheric aerosols (Dudeja, 2019; Martucci et al., 2007). A more complex vertical backscatter signal profile can also be formed under

specific environmental conditions, such as cloud contamination and local signal noise effect (Dang et al., 2019a; Stull, 1988).

The classical methodologies of lidar-retrieved algorithms are difficult to employ in the identification of multilayer structures in cases of night-time pollution. Gradient-based methods, such as the first-order gradient method (GM) (Hayden et al., 1997), inflexion point method (Menut et al., 1999), logarithm gradient method (Toledo et al., 2017), and cubic root gradient method (CRGM) (Yang et al., 2017), are sensitive to noisy data unless signal averaging is performed to prevent the

loss of some useful instantaneous information. The threshold method is too subjective to set a universal threshold for different weather and terrains (Frioud et al., 2003), while the variance method (Hooper and Eloranta, 1986) is easily affected by lofted aerosol layers and reduces the temporal resolution by calculating the variance profile. The Haar wavelet covariance transform (WCT) (Davis et al., 2000) and the idealised backscatter (Steyn et al., 1999) methods are more robust to noise; however, they can still be affected by low-level clouds and lofted aerosol layers (Caicedo et al., 2017). Some graph theory

methods, such as the extended Kalman filter (Banks et al., 2014), Pathfinder and PathfinderTURB (de Bruine et al., 2017; Poltera et al., 2017), $k$-means clustering (Liu et al., 2018; Toledo et al., 2014), and The STRAT-2D algorithm (Haeffelin et al., 2012) have been proposed to yield promising results via an automated method that reduces the incorrect detection of ABLH. However, these techniques strongly depend on the vertical distribution of particle layers (aerosols and clouds) and are prone to increase the uncertainty under complicated multilayer conditions.

The retrieval of ABLH under cloudy conditions is quite challenging. Some researchers have used the threshold of the attenuated scattering ratio (Campbell et al., 2008; Winker and Vaughan, 1994), the ratio of peaks to the base of the range-corrected signal (RCS) (Wang and Sassen, 2001) to locate cloud tops and bases, while others have employed the objective upper limit of the convective condensation level (CCL)(Li et al., 2017a), as well as analyzed the signal continuity and classify whether the cloud caps within the ABL (Dang et al., 2019b). The height restriction has significant advantages in

removing the influence of clouds. Elevated aerosol layers (EALs) are characteristically similar to the aerosol trapped in ABL, using the threshold of lidar backscatter coefficient can distinguish them (Dubovik et al., 2002; Hänel et al., 2012; Peng et al., 2017). More instrument and multi-wavelength lidar systems are combined to obtain more accurate results to identified the EALs (Liu et al., 2019; Ortega et al., 2016).

Digressing from these previous efforts to estimate the ABLH, we herein present a new approach—cluster analysis of the gradient method (CA-GM)—to overcome the multilayer structure and remove the noise fluctuation of NBLH with raw data resolution. This study proposes a reasonable parameter to reduce the interference of the cloud layer, EALs, and local noise over the air pollution in megacity regions. The results were evaluated by comparison with the nearby radiosonde site, and they were confirmed through continuous observation via traditional methods in different atmospheric layers.

## 2 Instruments and datasets

### 2.1 PM$_{2.5}$ data and Radiosonde

Beijing, located in the North Plain of China, experienced severe intermittent haze pollution from December 2016 to December 2017. The 39-d lidar and radiosonde data were recorded during that period, and the average concentrations of PM$_{2.5}$ reached 140 μg/m$^3$. The dataset for lidar, average PM, and air quality index are provided in Section 1 of the Supplemental Materials. In situ PM$_{2.5}$ daily measurements in China are primarily obtained from the official website of the China National Environmental Monitoring Centre (CEMC; http://cnemc.cn/). The radiosonde data are released daily from Nanjiao Station (39.80° N, 116.47° E), which is located southeast of the Beijing Institute of Technology lidar (BIT-lidar) system (39.95° N, 116.32° E). The L-band radiosonde provided a high-resolution profile of temperature, pressure, relative humidity, wind speed, and direction twice a day at 08:00 and 20:00 local standard time (LST) (Guo et al., 2016). The sample temporal resolution is 1.2 s (Zhang et al., 2018), and the vertical resolution is less than 20 m. Previous studies (Hennemuth and Lammert, 2006; Seidel et al., 2012) have adopted the radiosonde as a reference for detecting ABLHs for daily and annual changes in lidar measurements. We resampled the radiosonde data using linear interpolation to achieve the same vertical resolution of lidar and compared it to the 1-h average NBLH centred around the radiosonde launch times. As a result of the complexity of the transition during the morning and early at night, the boundary layer is in a transition between stable and unstable conditions. To determine NBLH from the radiosonde vertical profiles of temperature and humidity, the elevated temperature inversion layer or the height of a significant reduction in moisture is used (Peng et al., 2017). The potential temperature gradient (PTG) should have a good correlation with the relative humidity gradient (RHG), with an allowable error of 100 m (Wang and Wang, 2016). In this study, if the difference between the PTG and RHG is in excess of 100 m, the PTG is considered first, whereas if there is no significant temperature change or the evident changes belong to the cloud or EALs, the result of RHG is referred to as the NBLH.

### 2.2 BIT-lidar system

A single-wavelength Raman-Mie lidar is operated on the campus of the Beijing Institute of Technology, providing aerosol, cloud, ABLHs, and temperature measurements. This lidar system has been continuously enhanced to capture aerosol loading (Chen et al., 2014; Ji et al., 2018a). The standardised RCS is subjected to the correlation lidar factor correction (correction of electronic noise error, background noise error, and overlap factor) and distance correction. The backscatter coefficient can be

calculated using the Fernald method (Fernald, 1984), and the assumed lidar ratio is 70 sr (Rosati et al., 2016) owing to the polluted continental aerosol particles. The detailed parameters of the BIT-lidar are listed in Table 1.

**Table 1. Key parameters of the BIT-lidar**

| Parameter | BIT-lidar |
| --- | --- |
| Laser | Nd: YAG |
| Pulse energy | 180 mJ |
| Repetition | 20 Hz |
| Wavelength | 532 nm |
| Telescope | Newtonian |
| Telescope diameter | 0.4 m |
| Mode | coaxial |
| Temporal resolution | 50 s |
| Vertical resolution | 2.5 m |

## 3. Rationale and implementation of the novel algorithm

### 3.1 Weighted k-means clustering

The NBL shows more complex internal structure at night, the particulate can be used as an important indicator of atmospheric layering because its vertical distribution is strongly affected by the thermal and dynamic structure of the atmosphere (Neff and Coulter, 1986).The assumption of the NBL at which the aerosol concentration and turbulence intensity are significantly higher in the NBL top than in the free atmosphere (FA)(Dang et al., 2019a; Wang et al., 2020). Owing to the

influence of the multilayer structure, the minima RCS gradient are the potential locations of the NBL top. The assembly of these distinguishing peaks with height over time into groups can be considered as space- and time-averaged aerosol concentrations. Therefore, it can solve the inadvertent jump between different atmospheric layers. A theoretical schematic of the $k$-means clustering principle is shown in Figure 1. To form clusters, the Euclidean distance $dis(x_i, x_j)$ between two given signal points $x_i$ and $x_j$, with coordinates $(GM_i, h_i)$ and $(GM_j, h_j)$, is defined according to Eq. (1) as follows:

$$dis(x_i, x_j) = [(GM_i - GM_j)^2 + (h_i - h_j)^2)]^{1/2} \,, \tag{1}$$

where $GM_i$ $(GM_j)$ is the value of the RCS gradient and $h_i$ $(h_j)$ is the height of the peak.

Subsequently, we apply the $k$-means clustering algorithm to classify the datasets with the notable peaks. The cluster number is pre-set, and the $k$-means method builds clusters iteratively by moving the centroid until the target function sum of squared errors (SSE) approaches the local minimum (Toledo et al., 2014). The SSE is calculated using Eq. (2) as follows:

$$SSE = \sum_{i=1}^{k} \sum_{x \in Ci} dis(c_i, x)^2, \tag{2}$$

where $k$ is the number of clusters, $c_i$ and $x$ represent the cluster centroid and all observations in the cluster $Ci$, respectively. To obtain accurate data for compact and well-separated clusters, the criteria for cluster validation are necessary. The Davies–Bouldin index (Davies and Bouldin, 1979) is employed for the cluster validation analysis and defined as Eq. (3).

$$DB = \frac{1}{k}\sum_{i \neq j}^{k} max\left[\frac{S_k(c_i)+S_k(c_j)}{S(c_i,c_j)}\right],$$ (3)

where $S_k$ is the averaged intra-distance between the observations and their cluster centroid and $S(c_i, c_j)$ is the distance between cluster centroids $c_i$ and $c_j$. The minimum DB index is considered an optimal cluster classification.

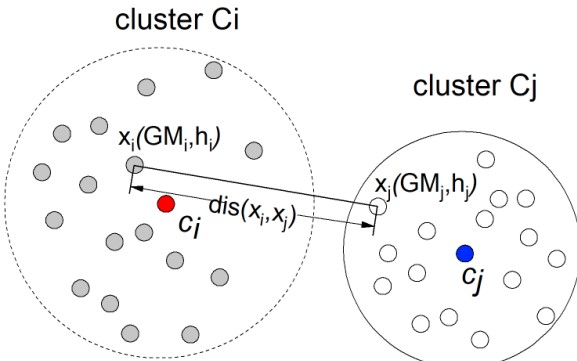

**Figure 1. Theoretical schematic of the k-means clustering. The Euclidean distance $dis(x_i, x_j)$ between two given signal points $x_i$ and $x_j$, with coordinates $(GM_i, h_i)$ and $(GM_j, h_j)$, in the cluster Ci and cluster Cj.**


The standard $k$-means algorithm must be normalised in cases where the variable is rather different, and data normalisation is based on min-max normalisation (Virmani et al., 2015). The normalised $k$-means clustering is 'isotropic' in all directions of space, and it tends to capture a spherical shape. Nevertheless, herein, we proposed to put weight on height and exclude variances greater along with height. Therefore, the assembling groups of the distribution tend to be separated along variables

with greater variances, which is conducive toward setting the upper and lower limiter altitude to classify different atmospheric layers vertically (Figure 2 b). The weighted $G$ is calculated by the difference between the maximum altitude $h_{max}$ and the minimum altitude, $h_{min}$, as shown in Eq.(4), while the weighted height $h_w$ is rescaled by the normalised height data $h_{nor}$, as presented in Eq.(5) as follows:

$$G = h_{max} - h_{min}, ,$$ (4)

$$h_w = G * h_{nor},$$ (5)

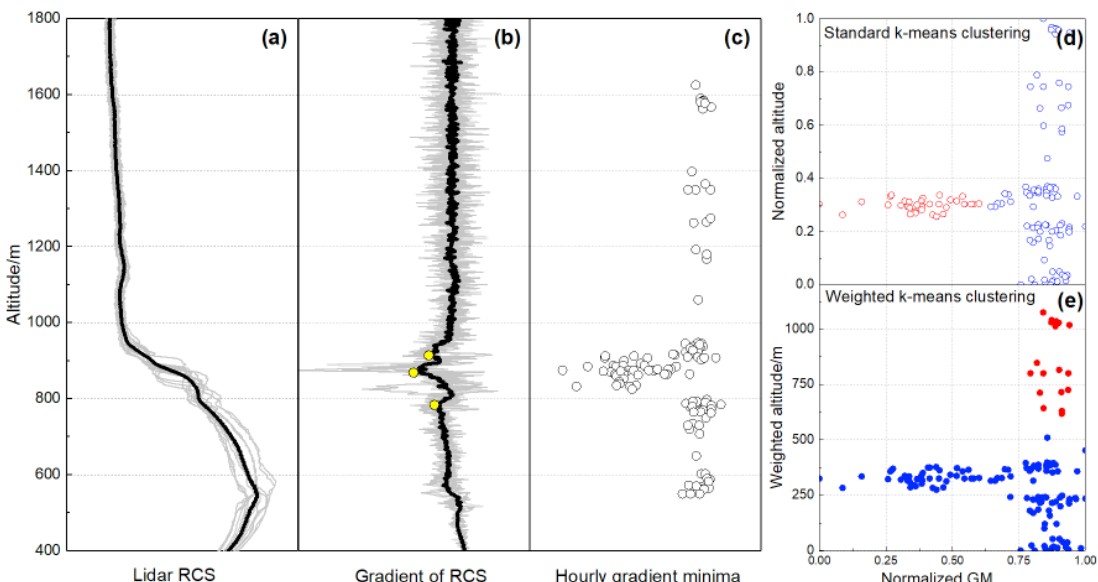

**Figure 2. The theoretical schematic of the weighted-*k* means clustering. (a)The real profile of a lidar RCS( light gray line) and the hour averaged RCS (black line). (b) The gradient of RCS (light gray line), the hour averaged gradient RCS (black line), and the three minima in the profile (yellow points). (c) The distribution of the gradient minima within an hour. (d-e) The results obtained by standard k-means and weighted k-means clustering, where two clusters are differentiated, as shown by red and blue hollow and solid points, respectively.**

## 3.2 Multilayer classification

Because of the presence of the strong gradient signature of the backscatter profile, a dataset of three minima of RCS gradient within an hour works as the dataset of k-means classification (Figure 2(e) ). The three minima are calculated by a window of 25 m, and selected in orders. The cloud layer (CL) has a larger gradient magnitude of extinction and backscattering coefficient than the aerosol layers (Palm et al., 2012). Additionally, the typical nocturnal clouds are shallow cumulus, stratocumulus, and stratus (Kotthaus and Grimmond, 2018). They have shallow vertical dimensions and are denser than aerosols at the same altitude; hence, they can be distinguished from the aerosol layer (Wang and Sassen, 2001). Meanwhile, the accuracy of the NBLH from GM can be affected by background and electronic noise; it has a non-regular distribution and appears at higher altitudes with lower signal-to-noise ratios. The noise layer lacks a stratified structure but has a GM value similar to that of the lower height. Thus, we calculated the range of vertical extension of different layers, indicating the cluster significance of the noise and other layers. As for the EALs, their presence above the NBL represents a difficulty when retrieving the upper height of the NBL, particularly when the EALs are close to it. Both aerosol layers have a similar characteristic of gradient variance and range of height, which we discover by seeking the empirical threshold value of the EALs in the backscatter coefficient (Hänel et al., 2012). The typical backscatter threshold for a 532-nm wavelength lidar is

defined as $\beta_{th} = 1.786 \times 10^{-3} \, km^{-1}sr^{-1}$, which is calculated using the Ångstöm parameter as 1.2 under urban-industrial and mixed conditions (Dubovik et al., 2002). The gaps between NBL and EALs in the multilayer structure are determined by $D_{th} = 100 \, m$ (Peng et al., 2017).

## 3.3 Implementation of the CA-GM algorithm

The CA-GM method, which is based on the $k$-means clustering analysis of different types of atmospheric layers, is generally used to retrieve multiple layers in polluted cases. The specific ideas are shown in the flowchart in Figure 3, and the specific steps are as follows: (Detailed results are presented as a case study.)

The algorithm is divided into three parts: pre-processing, layer attribution classification, and NBL inspection. The CA-GM algorithm is implemented if the data collection exceeds 30 min within an hour period. First, the standardised lidar RCS is applied to a Savitzky–Golay filter for preliminary denoising. The profile of the backscatter coefficient ($\beta$) is calculated, and the reference height ($h_{ref}$) is limited by the Fernald method as the theoretical height limiter (Comerón et al., 2017; Ji et al., 2017). Notably, $G^*$ is a dataset of three gradient minima of the RCS. The cluster is pre-set as a pair, and $k$-means clustering is carried out once to seek the minimum DB index as the optimal grouping, $Ci$ and $Cj$. Second, there is a parameter $D_{intra}$ that is defined as the minimum inter-cluster distance, which can measure the cluster stratified significance to classify the cloud and noise mixed in $G^*$. If $D_{intra}$ exceeds the threshold $D_{sig}$, it can distinguish the noise from other layers. $D_{sig}$ is the empirical value to distinguish noise layer for verified starfield. Furthermore, $S_{Ci}$ and $S_{Cj}$ are a quality control function for noise layer attribution. For the cloud layer, the vertical extension $R_{h_{Ci}}$ ($R_{h_{Cj}}$) of the cloud is lower than the aerosol layers; therefore, we define an empirical constant $C$ for this study. In addition, the vertical uniformity parameter $Vi$ ($Vj$) works as a quality control tool for the features of the cloud and other layers. If a cloud layer or noise exists, the original $G^*$ is removed from the upper limiter as $h_{cloud}$ or $h_{noise}$, respectively. After the elimination of cloud and noise interference, the EALs can be determined from the typical aerosol layer, $\beta_{th}$, and the gap distance, $D_{th}$. Finally, the new dataset, $G^*$, which has been removed from the different attributed layer, goes to the final step with the cluster as $Ci$ and $Cj$ (or $Ci'$ and $Cj'$). Owing to the assumption of NBL distribution, the largest deviation of cluster indicates the location of the NBLH ($h_{nbl}$).

In summary, by $k$-means clustering analysis of the vertical-temporal gradient of the GM once or twice within an hour, the multilayer NBL structure can be separated according to the physical characteristics of its different layers. The CA-GM method is an objective and robust method for judging the attribution of different layers (NBL, EALs, and CL) and noise.

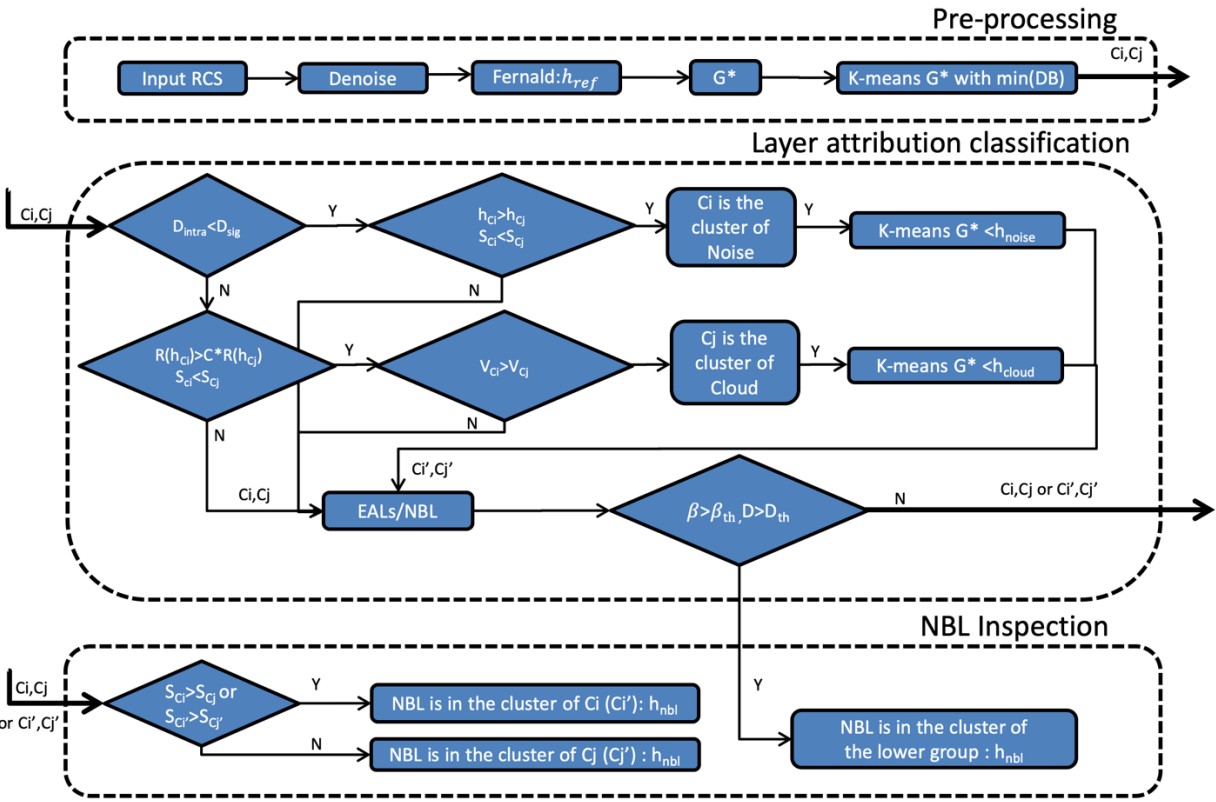

**Figure 3.** Flowchart of the retrieval method for the CA-GM; $h_{Ci}(h_{Cj})$: the height of centroid of cluster $Ci(Cj)$; $D_{intra}$: the inter-cluster distance between minimum $h_{Ci}$ and maximum $h_{Cj}$; $R_{h_{ci}}$: the intra-cluster range from the minimum $h_{Ci}$ and maximum $h_{Ci}$; $S_{Ci}(S_{Cj})$: the standard deviation of the GM in the cluster $Ci(Cj)$; $Vi(Vj)$: the vertical uniformity calculated by $R_{Ci} / N_{Ci}$ ($R_{Cj} / N_{Cj}$), the $N_{Ci}(N_{Cj})$ is the amount of peak in the group $Ci(Cj)$; $\beta_{th}$: typical backscatter aerosol layer ($1.786 \times 10^{-3} km^{-1} sr^{-1}$); $D_{th}$: threshold of distance to defined a gap between multiple aerosol layers (100 m); $D_{sig}$: empirical threshold as 50 m; $C$: empirical value as 1.5; and $h_{nbl}$: the final location of nocturnal boundary layer height.

## 4. Evaluation and comparative analysis with classical methods

### 4.1 Evaluation with radiosonde data

The L-band radiosonde provided accurate thermodynamic profiles, and the radiosonde-determined NBLHs were used to evaluate the accuracy of the lidar-retrieved NBLHs. Compared with the two-moment radiosonde with the other three algorithm, it was found that the correlation coefficients (R) ranged from 0.68–0.85. The CA-GM had the highest consistency among the classical methods, with the highest correlation coefficient (0.85), the weakest root mean square error (RMSE) (203 m), the smaller mean bias (28 m), and the minimum mean relative absolute difference (PRD) (17%) (Table 2).

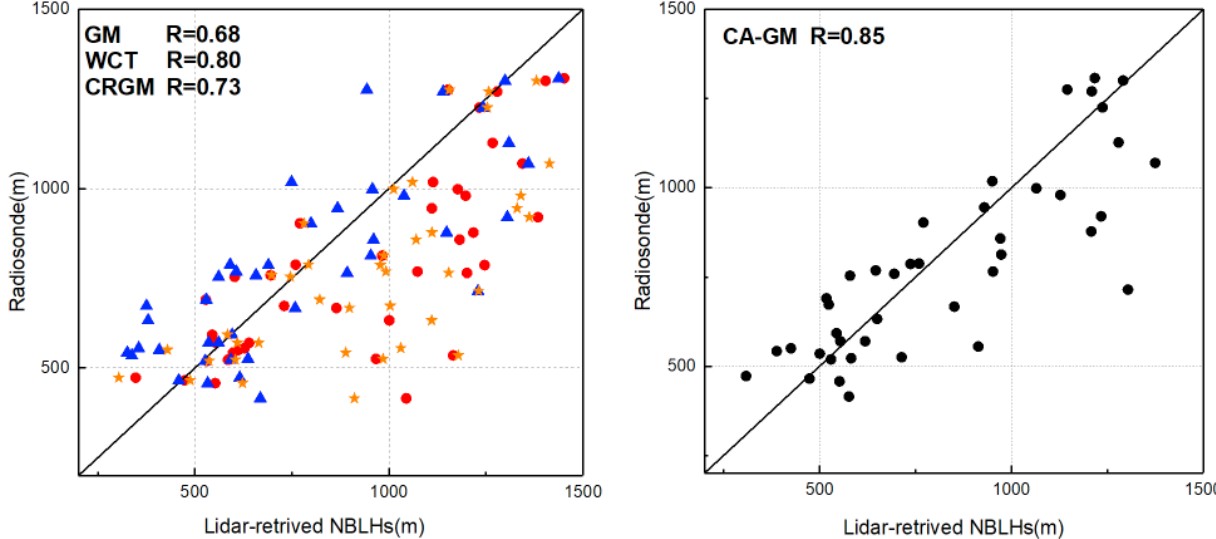


**Figure 4. Comparison between the radiosonde-determined and lidar-retrieved NBLH measurements via the gradient method (GM, red circle), wavelet covariance transform transition method (WCT, blue triangle), cubic root gradient method (CRGM, orange star), and cluster analysis of gradient method (CA-GM, black circle). The correlation coefficient is represented by R. The black solid line is the 1:1 line.**


The NBLH retrieved by GM and CA-GM (Figure 4) had a good correlation with the radiosonde approach, and the latter method enhanced the correlation coefficient by 25%. With the implementation of CA-GM, the data were concentrated, and the RMSE was reduced from 292–203 m (Table 2). The means bias of GM is greater than that of the CA-GM, corresponding to the decrease in PRD from GM to CA-GM. Additionally, compared with the WCT and CRGM, the former underestimated

the NBLH by approximately 13 m, whereas the latter overestimated the altitude by 186 m. The RMSE of CA-GM is less than that of WCT and CRGM, which is similar to the PRD result. Therefore, the CA-GM showed a good correlation with the radiosonde method, and evinced the least fluctuation and highest consistency in NBLH retrieval.

**Table 2. Statistic parameters of the lidar-retrieved algorithm compared with radiosonde measurement. Mean bias (MB),**
**correlation coefficient (R), root-means-square deviation (RMSE), and the percent of relative absolute bias different (PRD) are**
**shown below.**

| NBLH retrieved method | Mean Bias (m) | R | RMSE (m) | PRD (%) |
|---|---|---|---|---|
| Gradient method (GM) | 162 | 0.68 | 292 | 30 |

| | | | | |
|---|---|---|---|---|
| Wavelet covariance transform transition method (WCT) | -13 | 0.80 | 241 | 21 |
| Cubic root gradient method (CRGM) | 186 | 0.73 | 277 | 32 |
| Cluster analysis with gradient method (CA-GM) | 28 | 0.85 | 203 | 17 |

## 4.2. Comparison with other classical methods

To enrich our analysis, a comparison of CA-GM with GM, WCT, and CRGM in the 39-day night-time period was applied to compensate for the rare temporal resolution of the radiosonde approach. The results are shown in Figure 5.


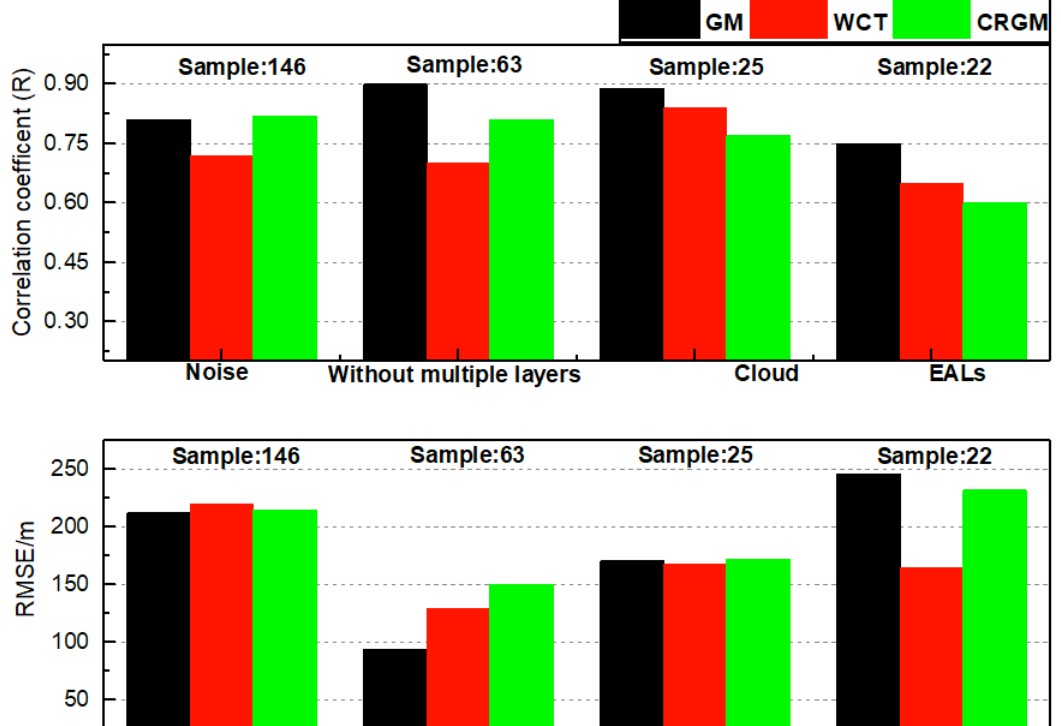

**Figure 5. Correlation coefficient and RMSE results compared with the CA-GM method under all conditions (see text for details) using the gradient method (GM), wavelet covariance transform transition method (WCT), and cubic root gradient method (CRGM). The sample number is shown at the top of the column, and the condition is represented by the x-axis.**


Valid CA-GM data were implemented for a total of 256 h, and the data were analysed for comparison with other retrieval algorithms. Under the condition without the infatuations of multiple layers, the CA-GM had a good correlation of 0.90, 0.70, and 0.82 for GM, WCT, and CRGM, respectively, and the RMSE was the least compared with other situations.

Consequently, the CA-GM was more similar to the other three methods in the case without the multilayer structure, which
proved the feasibility of the CA-GM relative to the classical boundary layer retrieval methods.

Moreover, the extensive results showed that the WCT method was more accurate than the GM during the night (Caicedo et al., 2017), and it was less affected by the low signal-to-noise ratio condition (Brooks, 2003). The dilation and threshold of the WCT method were selected carefully in this study (Mao et al., 2013); thus, the performance of the WCT could ensure the identification of the noise and most of the cloud layers. Notably, compared with the consistency for the WCT to the CA-GM,
the improvement of the correlation coefficient from 0.70 to 0.84 in cloud contamination and from 0.70 to 0.72 in noise effect was observed, which prove the ability to remove the attributed layers. Although the fluctuations in noise and cloud layers were relatively large, the CA-GM exhibited an outstanding ability for cloud removal to eliminate noise. As for EALs, because of their ambiguous cluster, as well as the NBL, all the methods had poor correlation coefficients with the CA-GM. Observing EALs is the most challenging part in multilayer structures; hence, more active remote sensing instruments (such
as multi-wavelength lidar and polarised lidar), as well as methods are required to determine the accurate layout of EALs.

Table 3 presents the criterion parameters in the CA-GM. The cluster significant parameter $D_{intra}$ for noise was 20.75 ± 14.62 m, which was significantly less compared to other conditions. The typical altitude of NBL, EALs, cloud, and noise in severe haze pollution is 590.49 ± 202.84 m, 1024.69 ± 166.36 m, 1252.52 ± 303.28 m, and 1100.66 ± 253.04 m, respectively. The vertical extension of the cloud layer was shallower than the other layer, with a typical extension of 128.6 ± 82.13 m. The
backscatter coefficient of EALs was 1.12 ± 0.76 × 10$^{-3}$ km$^{-1}$sr$^{-1}$, which was an evidence of choosing a suitable empirical $\beta_{th}$ value. The cloud had the smallest value in vertical uniformity, which indicated a denser peak distribution than other layers.

**Table 3. Computed criteria parameters for layer attribution**

| Parameter definition | Parameter | NBL | EALs | Cloud | Noise |
|---|---|---|---|---|---|
| Cluster signification (m) | $D_{intra}$ | 119.84 ± 83.70 | 103.41 ± 87.41 | 198.3 ± 86.69 | 20.75 ± 14.62 |
| Altitude (m) | $h_{Ci}, h_{Cj}$ | 590.49 ± 202.84 | 1024.69 ± 166.36 | 1252.52 ± 303.28 | 1100.66 ± 253.04 |
| Vertical extension (m) | $R_{h_{Ci}}, R_{h_{Cj}}$ | 383.77 ± 188.02 | 317.39 ± 89.59 | 128.6 ± 82.13 | 390.14 ± 176.58 |
| Backscatter coefficient (km$^{-1}$sr$^{-1}$) | β | 6.23 ± 5.36 × 10$^{-3}$ | 1.12 ± 0.76 × 10$^{-3}$ | 7.77 ± 7.42 × 10$^{-3}$ | 6.55 ± 8.40×10$^{-4}$ |
| Vertical uniformity | $Vi, Vj$ | 5.95 ± 2.19 | 5.87 ± 2.47 | 4.63 ± 1.63 | 7.39 ± 4.21 |

**4.3 Case study with a multilayer structure**

**4.3.1 Effects of cloud contamination**

On 23 December 2016, there was a cloud layer that was 1.3 km above ground level (AGL) between 18:00–23:00 LST (Figure 6-1), which was presented as a light blue region. Below the cloud base, there was a distinct aerosol layer surface and a strong signal negative gradient, indicating the WCT method capture. The cloud significantly influences the GM and CRGM determination and captures the upper edge of the cloud. After 21:00 LST, the cloudiness decreases, and the lidar can
capture the NBL signal. After defining the minimum in the upper cluster ($C_i$) as the top limiter altitude, the CA-GM captured

slowly increased the NBLH, as shown in Figure 6-1. Figure 6-2 shows the significant two-layer structure distribution hourly for the first *k*-means clustering distribution. The centroid of the two layers indicated the approximate location at 839 m and 1428 m (Figure 6-2 b), and the cloud located at the upper layer, which had a shallow vertical extension and a relatively dense distribution. The radiosonde measurement had a good correlation with the lidar-retrieved NBLH in Figure 6-3. The PTG
exhibited the steepest slope at 1.37 km, but it corresponded to the height at the cloud location. Therefore, we selected NBLH, using the RHG method, as 0.78 km, which was less than the CA-GM retrieved height at 20:00 LST.

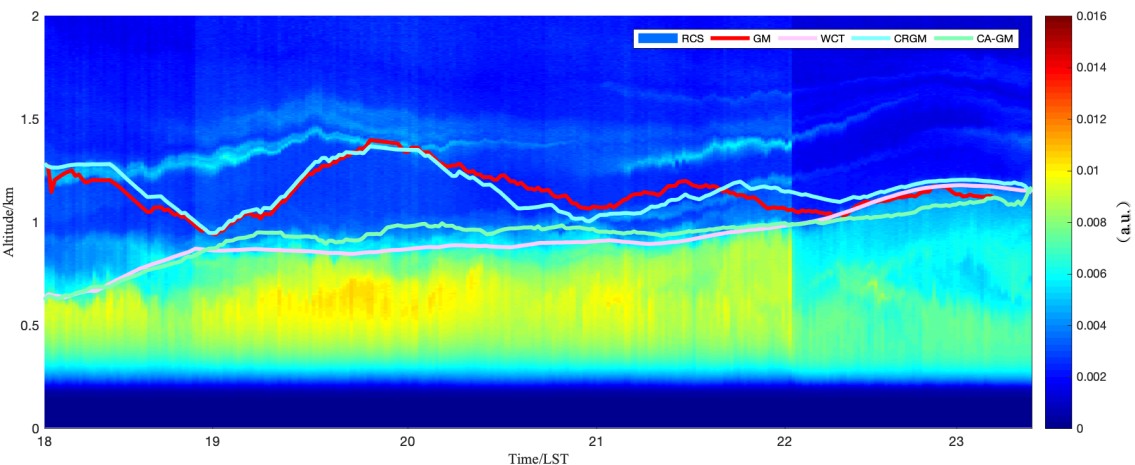

**Figure 6-1. Time–height cross section of range-corrected signal (RCS) with four NBLH retrieved methods on 23 December 2016.**

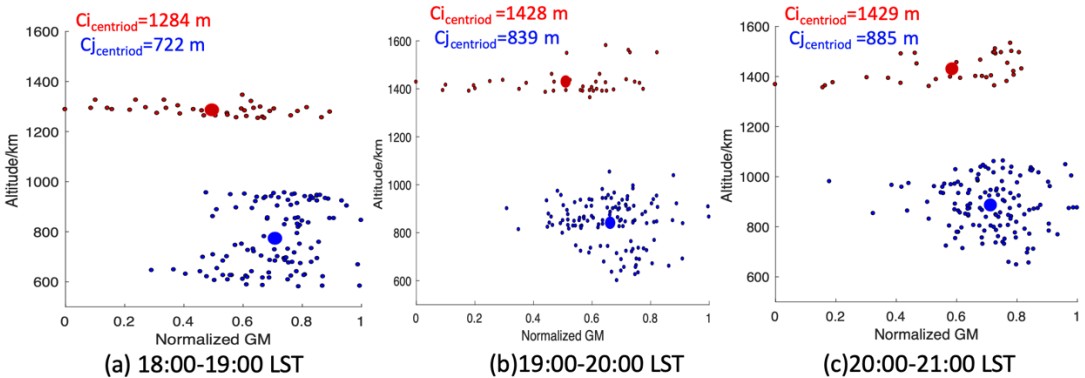

**Figure 6-2. Distribution of altitude and normalised gradient method (GM) values at 18:00–21:00 LST. (a), (b), and (c) indicate hourly intervals.**

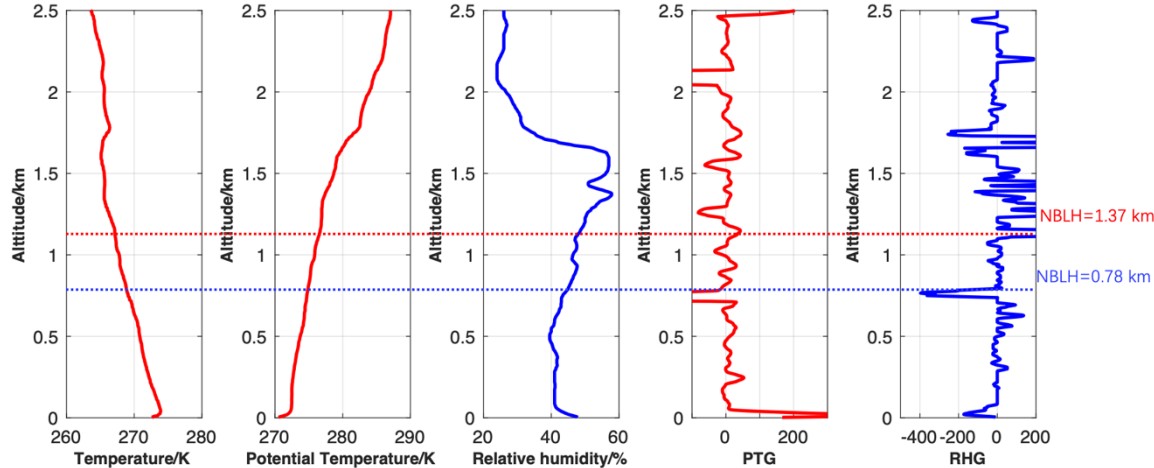

**Figure 6-3. Planetary boundary layer height estimates using radiosonde. Profiles include temperature, potential temperature, relative humidity, potential temperature gradient (PTG), and relative humidity gradient (RHG). Estimated NBLH by PTG (red) and RHG (blue) are shown by dashed horizontal lines.**

### 4.3.2 Noise effect

On 6 April 2017, the noise distribution was prone to appear when the low-load aerosol was utilised for the GM. The gradient-based methods were affected by noise and with a wide range of fluctuations (Figure 7-1). Conversely, the WCT adequately captured the edge of the aerosol concentration. From the distribution of the GM with height distribution; Figure 7-2 shows evident mixing without a stratified layer structure. Therefore, the noise was mixed in the upper layer of the centroid at 1479, 1452 at 19:00 and 20:00, which set the upper limiter and recalculated the NBLH in an hourly manner. Due to the standard deviation is not meet the requirement of the algorithm( $S_{ci} < S_{cj}$ ). Therefore, the NBL are in the cluster of the upper layer ($S_{ci}$=0.016, $S_{cj}$=0.033).The radiosonde data were calculated through the rapid change of the PTG method as 0.79 km (Figure 7-3), corresponding to the height retrieval by using CA-GM as 0.74 km.

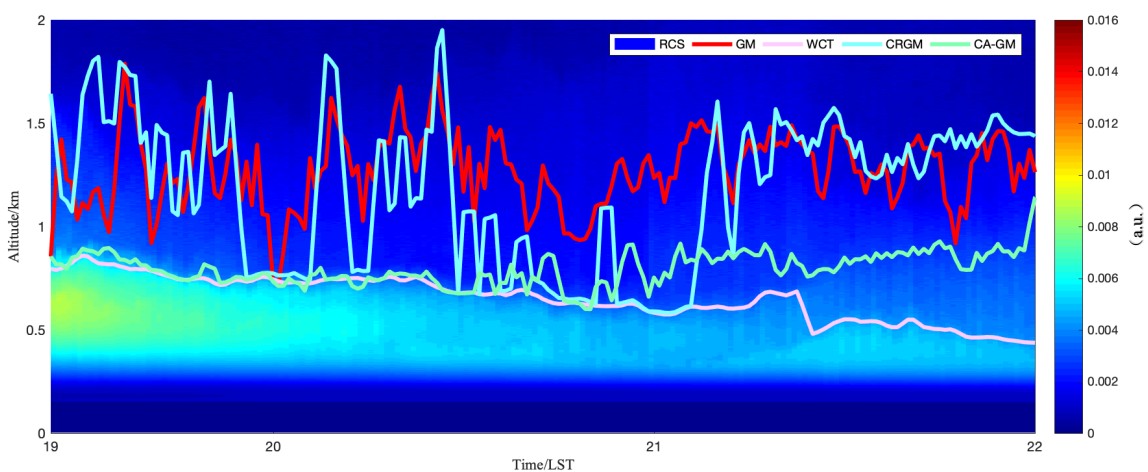

**Figure 7-1. Time–height cross section of RCS with four NBLH retrieval methods on 6 April 2017.**

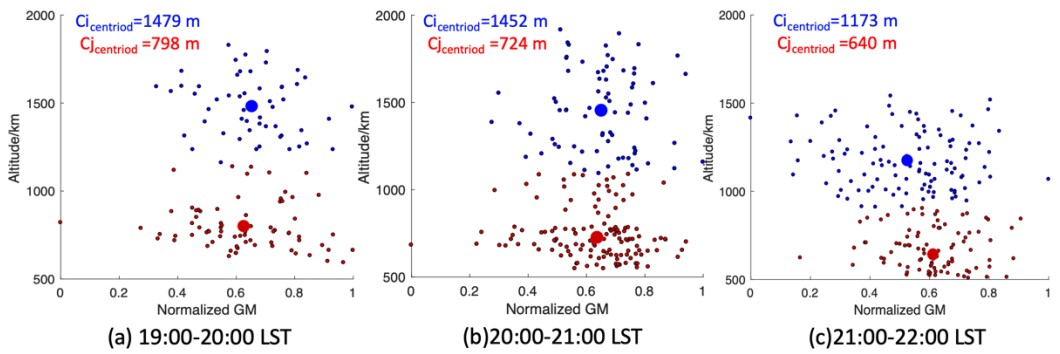

(a) 19:00-20:00 LST     (b)20:00-21:00 LST     (c)21:00-22:00 LST

**Figure 7-2. Distribution of altitude and the normalised gradient method (GM) value during 19:00–22:00 LST**

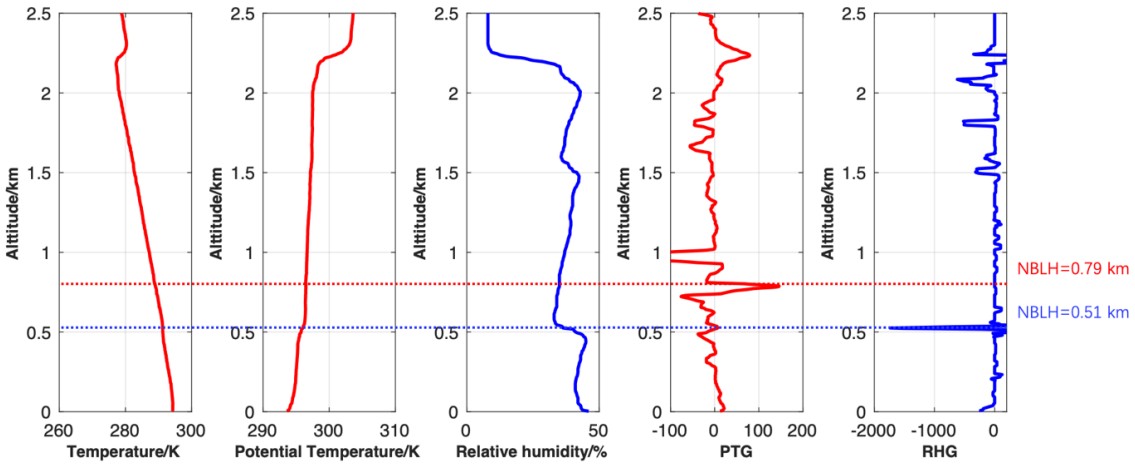

**Figure 7-3. Planetary boundary layer height estimates using radiosonde. Profiles include temperature, potential temperature, relative humidity, potential temperature gradient (PTG), and relative humidity gradient (RHG). Estimated NBLH by PTG (red) and RHG (blue) are shown by dashed horizontal lines at 20:00 LST**

### 4.3.3. Nocturnal aloft aerosol layer

On 2 January 2017, the EALs appeared frequently in the lower troposphere. There was a distinct aerosol layer between 0.7–

1.2 km AGL between 17:00–22:00 LST (Figure. 8-1). Without any limitation, the GM, CRGM, and WCT captured the height of the EALs when the negative gradient signal at the EALs was stronger than the NBL, corresponding to the lofted aerosol structure from 17:00–22:00 LST. As shown in Figure 8-2, the two distinct peaks of the cluster were the two aerosol layers; the deviation of the upper layer was larger initially and both layers gradually exhibited approximately the same gradient magnitude as the time transition. The upward centroid of the upper cluster provides additional evidence for the

NBLH with topped EALs. After using the CA-GM method to limit the base of the lofting aerosol layers, the effect of EALs

in the polluted cases can be successfully separated. Similarly, in Figure 8-3, the first gradient maxima above the surface inversion layer is the NBLH, and both PTG and RHG showed good consistency, while the NBLH was at 0.44 km. The other peak with PTG and RHG corresponded to the height of the EALs.

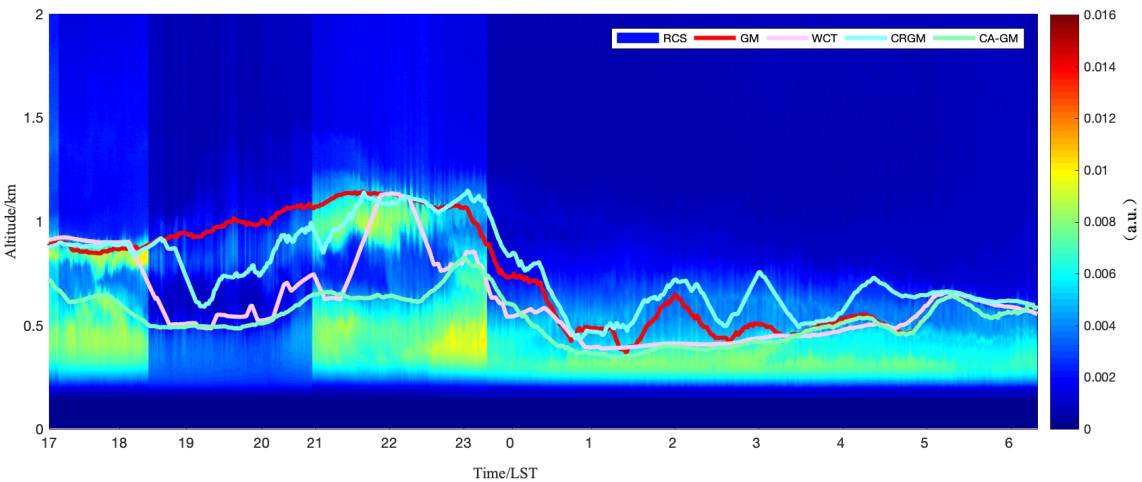


**Figure 8-1. Time–height cross section of RCS with four NBLH retrieval methods.**

(The discontinuity of the RCS at 18:06–18:07 is the result of detecting electric noise. The discontinuities of RCS at 20:39–20:58 and 23:18–23:39 were because of the laser energy adjustment and the signal test.)

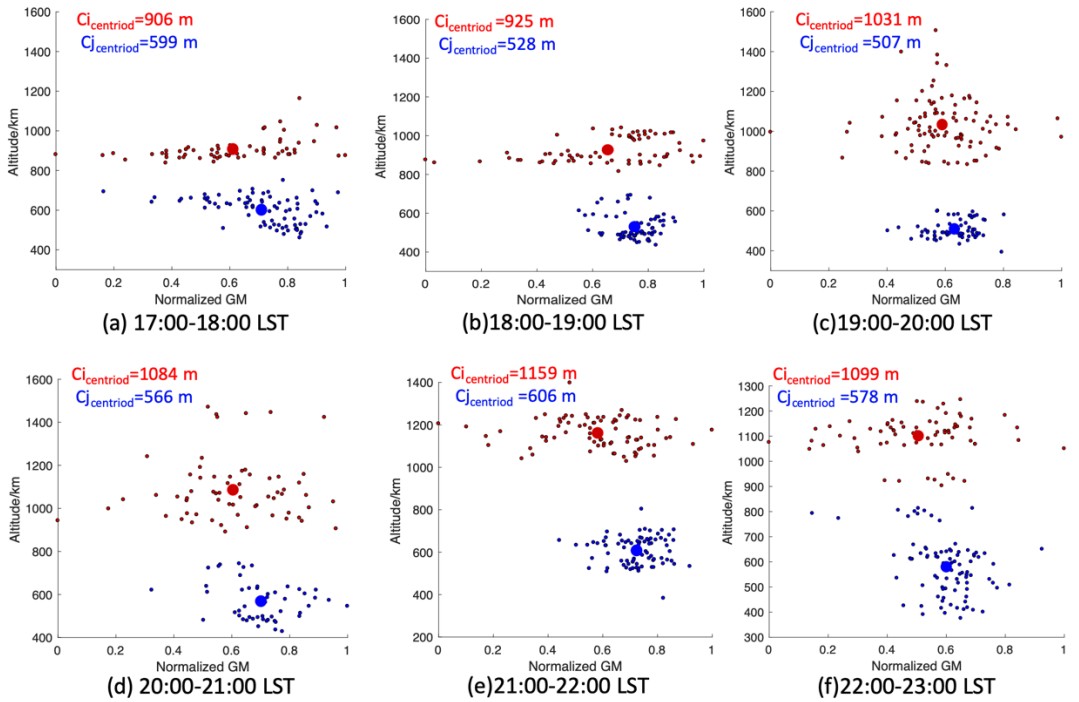

**Figure 8-2. Distribution of altitude and the normalised gradient method (GM) value during 16:00–23:00 LST.**

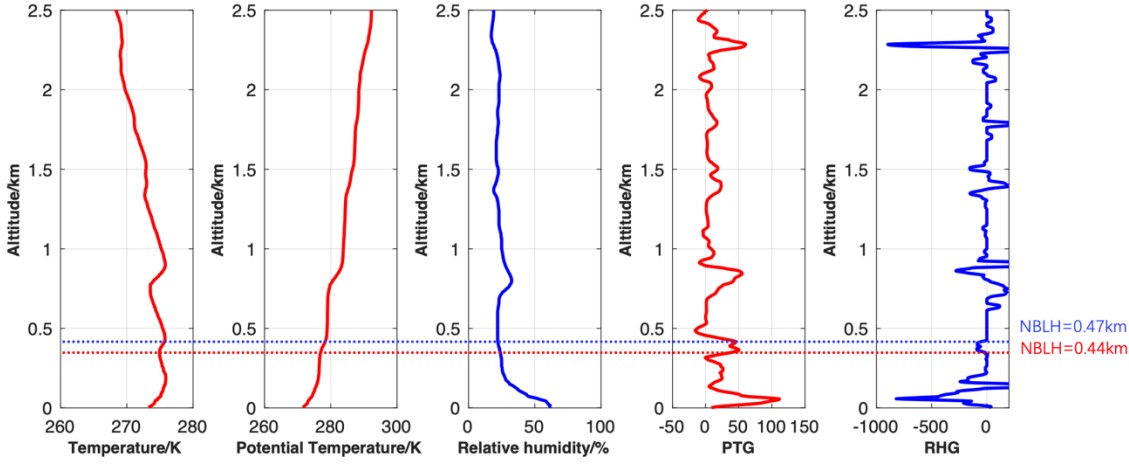


**Figure 8-3. Planetary boundary layer height estimates using radiosonde. Profiles include temperature, potential temperature, relative humidity, potential temperature gradient (PTG), and relative humidity gradient (RHG). Estimated NBLH by PTG (red) and RHG (blue) are shown by dashed horizontal lines at 20:00.**

## 5. Discussion and conclusion

Elastic lidars are excellent instruments to determine the NBLH with high space and vertical resolutions. Multilayer structures in severely polluted cases impede buoyancy forces and influencing pollutant dispersion and dilution. Herein, a novel CA-GM algorithm was developed to capture the multilayer structure and achieve stability at night with raw resolution. A 39-d heavily polluted observation experiment over Beijing (China) thoroughly established the limitations of the current methods employed for boundary layer height determination; a suitable algorithm for pollution conditions was developed.

Overall, the CA-GM method highlights its high performance relative to the radiosonde approach; the best correlation (0.85), weakest RMSE (203 m), and an improved 25% correlation coefficient of the GM was established. The possible deviations are due to the different definitions of thermodynamic NBLs from radiosondes and aerosol NBLs. The sound data are also multi-layered because of the effect of the aerosol and cloud layers, and the radiosonde-retrieved NBLH combine the PTG and RHG methods to discuss the uncertainty of NBLs in the pollution period. The calculation of the three minimum

gradients can be used to determine the potential stratified layer structure, which provides a worst case scenario for estimating the surface concentrations of pollutants released into the NBL. Compared with the 39-d performance of other algorithms, a reasonable parameter selection can distinguish different atmospheric layers, such as cloud layer, elevated (or advected) aerosol, and random noise. The $D_{intra}$, $R_{h_C}$, $\beta$, and $Vi$ provide a novel idea of classifying multiple layers based on their physical characteristics, which is more objective for automatic clustering under complex conditions.

The correlation coefficient with the CA-GM and WCT had an elevated correlation coefficient from 0.7–0.84 and 0.7–0.72 in cloud and noise effect, which proved the ability of the CA-GM to ensure the upper edge of the low-level cloud and remove the random noise. The EALs are often located at the top of NBLs with a similar characteristic of the NBLs. Thus,

using the empirical threshold on a single-wavelength elastic-lidar is a good way to classify EALs in polluted cases. Consequently, the CA-GM approach can deal with the uncertainty of the multi-layered structure and obtain a stable NBLH with a high temporal resolution, which is expected to contribute to the long-term observation of the single-wavelength lidar system and micro pulse lidar monitoring in air pollution.

The resolution of the CA-GM is in a high resolution, comparing to the previous studies (Martucci et al., 2010; Su and Patrick McCormick, 2019; Tsaknakis et al., 2011).The averaged time for elastic lidar system are 2,15, or 30 minutes, it will lose the raw time resolution in tracing the aerosol distribution. However, the CA-GM are taking into account the overall set of observations in the effective time (at least exceed 30 min), and finally maintain the raw resolution of the data.

The uncertainty of the CA-GM is calculated by the real signal provided in Section 2 of the Supplemental Materials. Concerning the robustness of the CA-GM approach, the effect of the lidar RCS noise in determining the NBLH has been analyzed. Unlike other gradient-derivate methods, CA-GM results are slightly affected by lidar signal noise. NBL top height as obtained for 'noised' lidar RCS with additive gaussian noise coefficient $\alpha < 4\%$ is better than GM results. The intensity of the CLs changes $\pm 40\%$ will not affect with the classification of the CA-GM in the polluted cases, the significantly stratified structure is related to the relative difference on the backscatter signal. As for the EALs, the strict threshold will defined the EALs accurately. The limitation of the CA-GM is based on the nocturnal boundary layer is stable, hence, we can calculate the distribution of the minima RCS gradients in an hour interval to use weighted $k$-means clustering to work as height restriction to the layers. Secondly, based on the limitation of the lidar system. The lower limit of the BIT-lidar is around 300 m. Too shallow of nocturnal boundary layer height (NBLH) are not be detectable. Thirdly, the method should be used in the high SNR condition, such as night-time and air pollution.

*Code/Data availability*. Contact csy@bit.edu.cn for data requests.

*Author contributions*. Y.Z and S.C analysed the experimental data and co-wrote the paper. H.C and P.G. supported the experiment, as well as for the maintenance of BIT-lidar. S.Y.C designed and led the study. All co-authors discussed the results and commented on the manuscript.

*Competing interests*. The authors declare that they have no conflicts of interest.

*Acknowledgements*. We wish to thank the China Meteorological Administration (CMA) for providing the radiosonde data. This research was supported by the National Natural Science Foundation of China (No. 61505009).

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
