# Peer review of "A novel lidar gradient cluster analysis method of nocturnal boundary layer detection during air pollution episodes"

_Atmospheric Measurement Techniques, 2020_

## Referee Comment (RC1) · Anonymous Referee #1 · 8 Jul 2020

The article presented a new approach combining gradient method and cluster analysis to distinguish multi-layers (i.e., the cloud layer, the evaluated aerosol layer, and the noise layer) and therefore retrieving NBLH based on lidar data. More information about such layers can also be obtained by the K-mean cluster analysis. However, the writing of the article needs to be further improved. And some doubts about your work are as follows:

1) Figure 2 should be described clearly. Is the red solid line the lidar signal profile averaged every 1h in figure 2(a)? And I'm confused about the weighted altitudes in figure 2(b), is hw equals h (the real height) minus h$min$? If yes, the maximum of hw is obviously lower than 1000 m, why a point exist higher than 1000 m in your figure?

2) Line145: "a dataset of three gradient minima of RCS". Do your mean three gradient minima of RCS at every 50 s within 1h are chosen to have a k-means cluster analysis?

3) From table 3, the altitude of NBL is always lower than that of EALs, Cloud, and Noise layers, so is there a simple top limiter works?

4) The word "starfield" appears many times in the article. Do you mean "stratified"? Please confirm it.

5) From figure 7-2 (c), the NBLH between 21:00 to 22:00 LST is about 640 m (h$cjcenter$). However, from Figure 7-1, the NBLH of that time period is much higher, why?

6) In your CA-GM algorithm, the cluster number is set as two in prior, that is, except NBL, assume that there is only one layer exist above NBL. So what if two or more layers (EALs, cloud layer, or noise layer) exist above NBLH? Besides, I'm concerned that if there is no EALs, cloud layer, or noise layer, does the cluster method affect the NBLH retrieval? Are the NBLHs from the CA-GM similar to that from the GM?

7) Lines 40-53: the authors have clarified some BLH retrieval methods; however, more previous evaluation works should be cited here, for example: McGrath-Spangler et al., 2012, Li et al., 2017. McGrath-Spangler, E. L., and A. S. Denning (2012), Estimates of North American summertime planetary boundary layer depths derived from space-borne lidar, J. Geophys. Res., 117, D15101, doi: 10.1029/2012JD017615.
H Li, Y Yang*, X-M Hu, Z Huang, G Wang, B Zhang, T Zhang (2017), Evaluation of retrieval methods of daytime convective boundary layer height based on Lidar data, J. Geophy. Res. Atmos., 122, doi: 10.1002/2016JD025620.

Meanwhile, there are some studies have worked to detect cloud or aerosol layers based on lidar data, like Winker et al., 1994, Wang et al., 2001, Li et al., 2017, Dang et al., 2019, should also be cited here, and explain why your work is needed compared to the others'.
Winker, D.M.; Vaughan, M.A. Vertical distribution of clouds over Hampton, Virginia, observed by lidar under the ECLIPS and FIRE ETO programs. Atmos. Res., 1994, 34, 117–133.
Wang, Z.; Sassen, K. Cloud type and macrophysical property retrieval using multiple remote sensors. J. of Appl. Meteorol., 2001, 40, 1665–1683.
H Li, Y Yang*, X-M Hu, Z Huang, G Wang, B Zhang. Application of Convective Condensation Level Limiter in Convective Boundary Layer Height Retrieval Based on Lidar Data. Atmosphere, 2017, 8, 79, doi: 10.3390/atmos8040079.
Dang, R., Yang, Y., Li, H., Hu, X.-M., Wang, Z., Huang, Z., Zhou, T. and Zhang, T.: Atmosphere

Boundary Layer Height (ABLH) Determination under Multiple-Layer Conditions Using Micro-Pulse Lidar, Remote Sensing, 11(3), 263, doi:10.3390/rs11030263, 2019.

Also, some minor revisions as follows:

Line 23: Change "continues" to "continuous".

Line 30: Change "on observation" to "based on various observations".

Line 34: coefficient between what?

Lines 51-52: The sentence is difficult to under understand.

Line 55: The fluctuation of NBLH, such statement is not completed.

Line 63: Delete "in the experiment".

Line 87: the value of turbulence? Such statement is incorrect.

Line 127: Change "the noise from the GM" to "the NBLH from the GM".

Line 148: Please explain Dsig here.

Line 281: Change "influence " to "influencing".

---

## Referee Comment (RC2) · Anonymous Referee #2 · 17 Jul 2020

The paper entitled "A novel Mie lidar gradient cluster analysis method of nocturnal boundary layer detection during air pollution episodes" presents a novel method based on the cluster analysis of the gradient method to detect the nocturnal boundary layer from lidar signals. The method presented is of interest for the retrieval of the nocturnal boundary layer. However, the language and writing of the paper needs to be greatly improved. At its current state, the paper is quite confusing and difficult to follow. Regarding the content, consider the following general comments: Please, use more comprehensive and recent bibliography. Additionally, state clearly what is the advantage of the proposed method compared to previous studies. From your results, the improvement compared to previous methods is not so evident in some cases. Indicate

clearly what is the method used to retrieve the NBLH from the radiosonde data. In section 4.3 different criteria seems to be used to establish a reference NBLH for the comparison with the radiosonde without being clearly justified. In general, the discussion of your results needs to be improved. My suggestion is to resubmit the paper after the language has been carefully reviewed and the previous comments addressed.

———————————————————

---

## Referee Comment (RC3) · Anonymous Referee #3 · 11 Aug 2020

General comments:

The manuscript "A novel Mie lidar gradient cluster analysis method of nocturnal boundary layer detection during air pollution episodes" presented a new algorithm to retrieve the nocturnal boundary layer height (NBLH), based on cluster analysis of gradient method, using 39 days lidar observations. The radiosonde data were used to evaluate its performance of NBLH retrieval, and results show that the presented algorithm had a better agreement than the other 3 methods (GM, WCT, CRGM). A comparison of this new methods with the other 3 methods were also analysed and discussed, using a 256 hours data set.

[Figure]

The presented method is promising for improving the NBLH retrieval, and results look interesting. However, I don't think the current form can clearly deliver the information, and a number of point must be clarified. Major changes are needed, and the writing of the paper must be improved, before the manuscript can be considered for publication. Please see my comments below.

Specific comments:

1. The presented method can only be applied for the BIT-lidar or can be used for other elastic lidars? Why you used "Mie lidar" in the title? I think this method is not only valid for Mie lidar. The conditions/constraints of using such method should be discussed.

In this study, the raw data resolution is used to get a high time & vertical resolution, any comparison with other methods? Have you used any vertical smoothing? What is the final time & vertical resolution? If reader want to apply this method to another lidar system, what's the limitation? Some discussions are need.

2. No uncertainty/error study is presented. Such information should be added.

3. The description of methodology is not clear, please revise it.

L88, explain more about the assumption.

Fig2, add legend for red line, grey lines, colour circles etc. are GM peaks from the red line? More description needed.

L122, "three minima peaks", and L145 "three gradient minima", do they refer to the same information? Please clarify which minima criterion you used, the 3 minima gradient values of RCS? Or the peaks with minima values? You can also add these minima by the markers in figure 2.

L144, describe more about the reference height.

Fig4, you can put all other methods using different colour/marker.

[Figure]

L225, Are you sure it is a cloud layer? RCS looks very weak for this layer. It could be a lofted aerosol layer. If it is not a cloud layer, another case should be presented in this section.

Technical corrections:

L12, 39 days is not a "long-term", maybe another expression.

L32-33, rephrase the sentence.

L67, provide the vertical resolution of radiosonde.

L73, add "gradient" for PTG

L76, is BIT-lidar rotational Raman–Mie lidar, but in this study you only use the elastic channel?

L79, after the overlap correction, what's the lower limit for BIT-lidar?

L89, NBL top. Add "top" here.

L115, change hw to hnor

L127, "the noise . . . be affected" do you mean the accuracy can be affected.

L130, add "layers" for EALs.

L170, what do you means here "with all algorithm"?

L206, any value for this "low SNR condition"?

L208, please specify which "improvement".

L282, "was automatics developed"?

L283, "high time resolution", please specify it. is it equal to the lidar vertical resolution?

---

## Author Comment (AC1) · 15 Sep 2020

**Responds to Anonymous Referee #1:**

**General comments:**

The article presented a new approach combining gradient method and cluster analysis to distinguish multi-layers (i.e., the cloud layer, the elevated aerosol layer, and the noise layer) and therefore retrieving NBLH based on lidar data. More information about such layers can also be obtained by the K-mean cluster analysis. However, the writing of the article needs to be further improved. And some doubts about your work are as follows:

**Response**:**

5

10

Thanks a lot for your reviews on our manuscript entitled "A novel Mie lidar gradient cluster analysis method of

nocturnal boundary layer detection during air pollution episodes (ID: amt-2020-167). We have revised the manuscript according to your suggestion, the language has been polished by Elsevier Language Editing Services and mentioned references have been added. The details are shown as follows.

**Specific comments:**

15 1.Figure 2 should be described clearly. Is the red solid line the lidar signal profile averaged every 1h in figure 2(a)? And I'm confused about the weighted altitudes in figure 2(b), is hw equals h (the real height) minus hmin? If yes, the maximum of hw is obviously lower than 1000 m, why a point exist higher than 1000 m in your figure?

**Response**:**

'Figure 2. The theoretical schematic of the weighted-k means clustering. (a)The real profile of a lidar RCS( light gray line) and the hour averaged RCS (black line). (b) The gradient of RCS (light gray line), the hour averaged gradient RCS (black line), and the three minima in the profile (yellow points). (c) The distribution of the gradient minima within

25

30

an hour. (d-e) The results obtained by standard k-means and weighted k-means clustering, where two clusters are differentiated, as shown by red and blue hollow and solid points, respectively.'

2) Yes, the weight in k-means clustering equals 1075 ( $G = h_{max} - h_{min}$ ). We modified the scale of the y-axis range, and check that there are several points located at around 1600 m, which indicated the weighted k-means points larger than 1000 m.

Thank you for your suggestion. The figure 2 has been changed.

2. Line145: "a dataset of three gradient minima of RCS". Do your mean three gradient minima of RCS at every 50 s within 1h are chosen to have a k-means cluster analysis?

**35 **Response:**

Yes, every profile of the RCS gradient is used to seek the three minima. Then, all the minima within an hour are used as the dataset of k-means classification.

The contents has been added to the article in P6 line 144.

'a dataset of three minima of RCS gradient within an hour works as the dataset of weighted k-means classification.'

45

50

3 From table 3, the altitude of NBL is always lower than that of EALs, Cloud, and Noise layers, so is there a simple top limiter works?

**Response:**

No, not exactly, the method contains height restriction on both upper limiter and lower limiter.

In this algorithm, it contain the top limiter conception in the first weighted k-means analysis, because the location of cloud layer and noise is above the NBL. Previous studies (Dang et al., 2019b; Li et al., 2017a) have successfully evaluate the works of the top-limiter.

However, in the second weighted k-means cluster processing. In order to classify the elevated aerosol layers (EALs) and NBL, we use the distance between two aerosol layers and the threshold of the backscatter coefficient as a sign to identifying the EALs and NBL. Here is an example of the height restriction on 18 Dec 2016 on 2: 00-3: 00 LST (Figure R1-1).

Through the first k-means clustering (Figure R1-1(a)), the noise is identified above 912.5 m. Next, in the second clustering analysis (Figure R1(b)), the upper groups are not meet the criteria for the EALs. After checking the standard

deviation of the normalized gradient method value between the centriod on each clusters ( $S_{Ci'} > S_{Cj'}$ ), we found that the NBL is the cluster of Ci'. The lower and top limiter are shown in the Figure R1(b).

Figure R1-1. The two weighted k-means clustering on 18 Dec 2016 between 2: 00-3: 00 LST.

(a) The first weighted k-means clustering. The results are shown by red and blue hollow and solid points, and their the centroids
 are represented by larger points of the same colour. The red dash line is shown the height of the limitation of noise. (b) The second weighted k-means clustering. The SCi and SCj represents the standard deviation of the normalized gradient values in each cluster. The two dash red line is the height restriction of the cluster.

4 The word "starfield" appears many times in the article. Do you mean "stratified"? Please confirm it.

**65 Response:**

Thank you for your suggestion. Yes, the word has been changed as stratified. (Line 151 & 170& 274).

5. From figure 7-2 (c), the NBLH between 21:00 to 22:00 LST is about 640 m (hcjcenter). However, from Figure 7-1, the NBLH of that time period is much higher, why?

**70 Response:**

In our algorithm, we have defined two constraints to identifying noise as in figure 3 shown. The first is that the noise signal distribution is not clearly stratified ( $D_{intra} < D_{sig}$ ), and the second is that the noise is located at a higher height( $h_{ci} > h_{cj}$ ) and the average standard deviation of the points in noise cluster is smaller than the NBL( $S_{ci}

---

## Author Comment (AC2) · 15 Sep 2020

**Responds to Anonymous Referee #2:**

The paper entitled "A novel Mie lidar gradient cluster analysis method of nocturnal boundary layer detection during air pollution episodes" presents a novel method based on the cluster analysis of the gradient method to detect the nocturnal boundary layer from lidar signals. The method presented is of interest for the retrieval of the nocturnal boundary layer. However, the language and writing of the paper needs to be greatly improved. At its current state, the paper is quite confusing and difficult to follow. In general, the discussion of your results needs to be improved. My suggestion is to resubmit the paper after the language has been carefully reviewed and the previous comments addressed.

**Response:**

Thanks a lot for your reviews on our manuscript entitled "A novel Mie lidar gradient cluster analysis method of nocturnal boundary layer detection during air pollution episodes (ID: amt-2020-167). We have revised the manuscript according to the comments, the language has been polished by Elsevier Language Editing Services. Moreover, the comprehensive reference and the discussion of the results have been added. The details are shown as follows.

**General comments:**

1. Please, use more comprehensive and recent bibliography.

**Response:**

The recent bibliography of boundary layer detection have been add in the P2 49-63.

The latest developments in the boundary layer height measurement, including classical methodology, graphic methodology, and algorithm with multiple layer structure interference have been added and express the relationship of our studies.

'

'*Some graph theory methods, such as the extended Kalman filter (Banks et al., 2014), Pathfinder and PathfinderTURB (de Bruine et al., 2017; Poltera et al., 2017), k-means clustering (Liu et al., 2018; Toledo et al., 2014), and The STRAT-2D algorithm (Haeffelin et al., 2012) have been proposed to yield promising results via an automated method that reduces the incorrect detection of ABLH. However, these techniques strongly depend on the vertical distribution of particle layers (aerosols and clouds) and are unsuitable for use under complicated multilayer conditions (Granados-Muñoz et al., 2012).*

*The retrieval of BLHs under cloudy conditions is quite challenging. Some researchers have used the threshold of the attenuated scattering ratio (Campbell et al., 2008; Winker and Vaughan, 1994), the ratio of peaks to the base of the range-corrected signal (RCS) (Wang and Sassen, 2001) to locate cloud tops and bases, while others have employed the objective upper limit of the convective condensation level (CCL)(Li et al., 2017), as well as the analysis of signal continuity and the classification of whether the cloud caps the ABLH or is decoupled from the ABL (Dang et al., 2019). The height restriction has significant advantages in removing the influence of clouds. Elevated aerosol layers (EALs) are characteristically similar to the aerosol trapped in ABL, using the threshold of lidar backscatter coefficient can distinguish them (Dubovik et al., 2002;*

*Hänel et al., 2012; Peng et al., 2017). More instrument and multi-wavelength lidar systems are combined to obtain more accurate results to identified the EALs (Liu et al., 2019; Ortega et al., 2016).*

2. State clearly what is the advantage of the proposed method compared to previous studies. From your results, the improvement compared to previous methods is not so evident in some cases.

**Response:**

Thank you for your suggestion. We come up with an algorithm based on cluster analysis of gradient method (CA-GM) in the nocturnal boundary layer (NBLH). Compared to the gradient-based methods such as GM and CRGM, it will be more robust in a nosier condition. With the test of the real signal, the CA-GM has better performance through polluted cases (Figure R2-1). Secondly, although wavelet covariance transform (WCT) method is robust in noise affection, it can still be affect by low- level cloud and the aloft aerosol layers. As figure 8-1 shows, the time period on 17:00-18:00 and 21:00-22:00, the CA-GM has significant ability to capture the NBL. Thirdly, according to the Table 2, the best correlation with radiosonde (0.85), the smallest RMSE (203 m) with radiosonde are shown the better performance of the CA-GM in capturing the NBL in polluted cases.

The testing with the real signal are shown below.

Use the RCS(z) signal, and randomly noised $RCS^{noised}(z)$ by the expression:

$$RCS^{noised}(z) = RCS(z) + [\alpha \times \chi(z)] \quad (R2\text{-}1)$$

Where $\chi(z)$ is the random noise function taking values between 0 and 1, z is the height, and $\alpha$ is a varying parameter as introduced in Eq (R2-1) to produces different levels of noise.

[Figure]

Figure R2-1. The real lidar RCS for the heavily polluted case (17 Dec 2016 20:00-21:00 LST ).
(a-c) three noise level cases, (d-f) with the gradient of RCS, and (g-i) the first weighted k-means clustering.

[Figure]

**Figure R2-2 RMSE between the WCT and the other three algorithms (GM,CRGM and CA-GM )**

60   As a result of the figure R2-2 shows, the CA-GM has less RMSE than GM at the ratio of 1%-4%. The figure R2-1 (g-h) shows similar groups in different range of noise affection. However, the clustering changes at the results of R2-1(i). Due to the noise distribution of the signal, the centriod of the cluster will get higher and lose the ability to restrict the changes of GM. The difference in the NBL top is found with the noise level 4% case, they are lower than 1% in respect to the estimate for the case with raw signal.

[Figure]

**Figure R2-3. The real lidar RCS for the cloud case (5 Jan 2016 00:00-1:00 LST).**

**(a-d) the different ratio of strength of the cloud layer intensity. (e-h) the first weighted-means clustering**

Add the signal of the cloud layer on the raw data, the ratio of the intensity for cloud layer changes from -40% to 40%. As the figure shown the first k-means clustering in figure R2-3(e-h),the intensity of the cloud layer will not influence the CA-GM.

In summary, these results indicate that the degree of estimation of the NBL top by applying CA is weakly affected by the signal noise. In fact,  a few NBLH depending on the value of the RCS gradient in a discrete point. CA determines the NBL by taking into account the overall set of observations of a given point, thus decreases the dependence of the method on the RCS values in single moment. The intensity of the CLs changes ±40% and will not affect the cluster of the CA-GM, it can be significant stratified due to the relative significantly signal difference on the backscatter signal. As for the EALs, the strict threshold will defined the EALs accurately. Therefore, the CA-GM approach is able to accurately obtain the NBLH with the effect of noise, cloud layers and elevated aerosol layers.

The superiority of the CA-GM are added at the discussion section.

' *In relation to the robustness of the CA-GM approach, the effect of the lidar RCS noise in determining the NBLH has been*

*analyzed. Unlike other gradient-derivate methods, CA-GM results are slightly affected by lidar signal noise. NBL top height as obtained for 'noised' lidar RCS with value of α < 4% is better than GM results. The intensity of the CLs changes ±40% will not affect with the cluster of the CA-GM in the polluted cases, the significantly starfield structure is related to the relative difference on the backscatter signal. As for the EALs, the strict threshold will defined the EALs accurately.'*

3. Indicate clearly what is the method used to retrieve the NBLH from the radiosonde data. In section 4.3 different criteria seems to be used to establish a reference NBLH for the comparison with the radiosonde without being clearly justified.

**Respond:**

The different criteria description of the radiosonde data was added in the Section 2.1.

*'As a result of the complexity of the transition during the morning and early at night, the boundary layer is in a transition between stable and unstable conditions. To determine NBLH from the radiosonde vertical profiles of temperature and humidity, the elevated temperature inversion layer or the height of a significant reduction in moisture is used (Peng et al., 2017). The potential temperature gradient (PTG) should have a good correlation with the relative humidity gradient (RHG), with an allowable error of 100 m (Wang and Wang, 2016). In this study, if the difference between the PTG and RHG is in*

*excess of 100 m, the PTG is considered first, whereas if there is no significant temperature change or the evident changes belong to the cloud or EALs, the result of RHG is referred to as the NBLH.'*

4. In general, the discussion of your results needs to be improved.

**Responds:**

Thank you for your suggestion. The discussion are add two parts of the content, about the uncertainty and the limitation of the algorithm. P17 Line 333-346.

*'The uncertainty of the CA-GM is calculated by the real signal. Concerning the robustness of the CA-GM approach, the effect of the lidar RCS noise in determining the NBLH has been analyzed. Unlike other gradient-derivate methods, CA-GM*

*results are slightly affected by lidar signal noise. NBL top height as obtained for 'noised' lidar RCS with value of α < 4% is better than GM results. The intensity of the CLs changes ±40% will not affect with the cluster of the CA-GM in the polluted cases, the significantly starfield structure is related to the relative difference on the backscatter signal. As for the EALs, the strict threshold will defined the EALs accurately. The limitation of the CA-GM is based on the assumption that the nocturnal boundary layer is stable, hence, we can calculate the distribution of the minima gradients of the RCS in an hour interval to*

*use weighted k-means clustering to work as height restriction to the layers. Secondly, based on the limitation of the lidar system. The lower limit of the BIT-lidar is around 300 m. Too shallow of nocturnal boundary layer height (NBLH) are not be detectable. Thirdly, the method should be used in the high SNR condition, such as night-time and air pollution. '*

Thank you so much for your reviewing! We deeply appreciate your recognition of our research work.

**Reference**

Banks, R. F., Tiana-Alsina, J., María Baldasano, J. and Rocadenbosch, F.: Retrieval of boundary layer height from lidar using extended Kalman filter approach, classic methods, and backtrajectory cluster analysis, edited by A. Comerón, E. I. Kassianov, K. Schäfer, R. H. Picard, K. Stein, and J. D. Gonglewski, p. 92420F, Amsterdam, Netherlands., 2014.

de Bruine, M., Apituley, A., Donovan, D. P., Klein Baltink, H. and de Haij, M. J.: Pathfinder: applying graph theory to consistent tracking of daytime mixed layer height with backscatter lidar, Atmospheric Measurement Techniques, 10(5), 1893–1909, doi:10.5194/amt-10-1893-2017, 2017.

Campbell, J. R., Sassen, K. and Welton, E. J.: Elevated Cloud and Aerosol Layer Retrievals from Micropulse Lidar Signal Profiles, Journal of Atmospheric and Oceanic Technology, 25(5), 685–700, doi:10.1175/2007JTECHA1034.1, 2008.

Dang, R., Yang, Y., Li, H., Hu, X.-M., Wang, Z., Huang, Z., Zhou, T. and Zhang, T.: Atmosphere Boundary Layer Height (ABLH) Determination under Multiple-Layer Conditions Using Micro-Pulse Lidar, Remote Sensing, 11(3), 263, doi:10.3390/rs11030263, 2019.

Dubovik, O., Holben, B., Eck, T. F., Smirnov, A., Kaufman, Y. J., King, M. D., Tanré, D. and Slutsker, I.: Variability of Absorption and Optical Properties of Key Aerosol Types Observed in Worldwide Locations, Journal of the Atmospheric

Sciences, 59(3), 590–608, doi:10.1175/1520-0469(2002)059<0590:VOAAOP>2.0.CO;2, 2002.

Granados-Muñoz, M. J., Navas-Guzmán, F., Bravo-Aranda, J. A., Guerrero-Rascado, J. L., Lyamani, H., Fernández-Gálvez, J. and Alados-Arboledas, L.: Automatic determination of the planetary boundary layer height using lidar: One-year analysis over southeastern Spain: DETERMINATION OF THE PBL HEIGHT, Journal of Geophysical Research: Atmospheres, 117(D18), n/a-n/a, doi:10.1029/2012JD017524, 2012.

Haeffelin, M., Angelini, F., Morille, Y., Martucci, G., Frey, S., Gobbi, G. P., Lolli, S., O'Dowd, C. D., Sauvage, L., Xueref-Rémy, I., Wastine, B. and Feist, D. G.: Evaluation of Mixing-Height Retrievals from Automatic Profiling Lidars and Ceilometers in View of Future Integrated Networks in Europe, Boundary-Layer Meteorology, 143(1), 49–75, doi:10.1007/s10546-011-9643-z, 2012.

Hänel, A., Baars, H., Althausen, D., Ansmann, A., Engelmann, R. and Sun, J. Y.: One-year aerosol profiling with EUCAARI

Raman lidar at Shangdianzi GAW station: Beijing plume and seasonal variations: ONE-YEAR AEROSOL PROFILING NEAR BEIJING, Journal of Geophysical Research: Atmospheres, 117(D13), n/a-n/a, doi:10.1029/2012JD017577, 2012.

Li, H., Yang, Y., Hu, X.-M., Huang, Z., Wang, G. and Zhang, B.: Application of Convective Condensation Level Limiter in Convective Boundary Layer Height Retrieval Based on Lidar Data, Atmosphere, 8(12), 79, doi:10.3390/atmos8040079, 2017.

Liu, B., Ma, Y., Liu, J., Gong, W., Wang, W. and Zhang, M.: Graphics algorithm for deriving atmospheric boundary layer heights from CALIPSO data, Atmospheric Measurement Techniques, 11(9), 5075–5085, doi:10.5194/amt-11-5075-2018, 2018.

Liu, B., Ma, Y., Gong, W., Zhang, M. and Yang, J.: Improved two-wavelength Lidar algorithm for retrieving atmospheric boundary layer height, Journal of Quantitative Spectroscopy and Radiative Transfer, 224, 55–61, doi:10.1016/j.jqsrt.2018.11.003, 2019.

Ortega, I., Berg, L. K., Ferrare, R. A., Hair, J. W., Hostetler, C. A. and Volkamer, R.: Elevated aerosol layers modify the $O_2$–$O_2$ absorption measured by ground-based MAX-DOAS, Journal of Quantitative Spectroscopy and Radiative Transfer, 176, 34–49, doi:10.1016/j.jqsrt.2016.02.021, 2016.

Peng, J., Grimmond, C. S. B., Fu, X., Chang, Y., Zhang, G., Guo, J., Tang, C., Gao, J., Xu, X. and Tan, J.: Ceilometer-Based

Analysis of Shanghai's Boundary Layer Height (under Rain- and Fog-Free Conditions), Journal of Atmospheric and Oceanic Technology, 34(4), 749–764, doi:10.1175/JTECH-D-16-0132.1, 2017.

Poltera, Y., Martucci, G., Collaud Coen, M., Hervo, M., Emmenegger, L., Henne, S., Brunner, D. and Haefele, A.: PathfinderTURB: an automatic boundary layer algorithm. Development, validation and application to study the impact on in situ measurements at the Jungfraujoch, Atmospheric Chemistry and Physics, 17(16), 10051–10070, doi:10.5194/acp-17-

10051-2017, 2017.

Toledo, D., Córdoba-Jabonero, C. and Gil-Ojeda, M.: Cluster Analysis: A New Approach Applied to Lidar Measurements for Atmospheric Boundary Layer Height Estimation, Journal of Atmospheric and Oceanic Technology, 31(2), 422–436, doi:10.1175/JTECH-D-12-00253.1, 2014.

Wang, X. and Wang, K.: Homogenized Variability of Radiosonde-Derived Atmospheric Boundary Layer Height over the

Global Land Surface from 1973 to 2014, Journal of Climate, 29(19), 6893–6908, doi:10.1175/JCLI-D-15-0766.1, 2016.

Wang, Z. and Sassen, K.: Cloud Type and Macrophysical Property Retrieval Using Multiple Remote Sensors, Journal of Applied Meteorology, 40(10), 1665–1682, doi:10.1175/1520-0450(2001)040<1665:CTAMPR>2.0.CO;2, 2001.

Winker, D. M. and Vaughan, M. A.: Vertical distribution of clouds over Hampton, Virginia observed by lidar under the ECLIPS and FIRE ETO programs, Atmospheric Research, 34(1–4), 117–133, doi:10.1016/0169-8095(94)90084-1, 1994.

---

## Author Comment (AC3) · 15 Sep 2020

**General comments:**

The article presented a new approach combining gradient method and cluster analysis to distinguish multi-layers (i.e., the cloud layer, the elevated aerosol layer, and the noise layer) and therefore retrieving NBLH based on lidar data. More
information about such layers can also be obtained by the K-mean cluster analysis. However, the writing of the article
needs to be further improved. And some doubts about your work are as follows:

**Response**:

Thanks a lot for your reviews on our manuscript entitled "A novel Mie lidar gradient cluster analysis method of nocturnal boundary layer detection during air pollution episodes (ID: amt-2020-167). We have revised the manuscript
according to your suggestion, the language has been polished by Elsevier Language Editing Services and mentioned
references have been added. The details are shown as follows.

**Specific comments:**

1.Figure 2 should be described clearly. Is the red solid line the lidar signal profile averaged every 1h in figure 2(a)? And
I'm confused about the weighted altitudes in figure 2(b), is hw equals h (the real height) minus hmin? If yes, the
maximum of hw is obviously lower than 1000 m, why a point exist higher than 1000 m in your figure?

**Response**:

1) The description of the figure 2 has been added at P6.line 138-142.

[Figure]

*'Figure 2. The theoretical schematic of the weighted-k means clustering. (a)The real profile of a lidar RCS( light gray line) and the hour averaged RCS (black line). (b) The gradient of RCS (light gray line), the hour averaged gradient RCS (black line), and the three minima in the profile (yellow points). (c) The distribution of the gradient minima within an hour. (d-e) The results obtained by standard k-means and weighted k-means clustering, where two clusters are differentiated, as shown by red and blue hollow and solid points, respectively.'*

2) Yes, the weight in k-means clustering equals 1075 ($G = h_{max} - h_{min}$). We modified the scale of the y-axis range, and check that there are several points located at around 1600 m, which indicated the weighted k-means points larger than 1000 m.

Thank you for your suggestion. The figure 2 has been changed.

2. Line145: "a dataset of three gradient minima of RCS". Do your mean three gradient minima of RCS at every 50 s within 1h are chosen to have a k-means cluster analysis?

**Response:**

Yes, every profile of the RCS gradient is used to seek the three minima. Then, all the minima within an hour are used as the dataset of k-means classification.

The contents has been added to the article in P6 line 144.

*'a dataset of three minima of RCS gradient within an hour works as the dataset of weighted k-means classification.'*

From table 3, the altitude of NBL is always lower than that of EALs, Cloud, and Noise layers, so is there a simple top limiter works?

**Response:**

No, not exactly, the method contains height restriction on both upper limiter and lower limiter.

In this algorithm, it contain the top limiter conception in the first weighted k-means analysis, because the location of cloud layer and noise is above the NBL. Previous studies (Dang et al., 2019b; Li et al., 2017a) have successfully evaluate the works of the top-limiter.

However, in the second weighted k-means cluster processing. In order to classify the elevated aerosol layers (EALs) and NBL, we use the distance between two aerosol layers and the threshold of the backscatter coefficient as a sign to identifying the EALs and NBL. Here is an example of the height restriction on 18 Dec 2016 on 2:00-3:00 LST (Figure R1-1).

Through the first k-means clustering (Figure R1-1(a)), the noise is identified above 912.5 m. Next, in the second clustering analysis (Figure R1(b)), the upper groups are not meet the criteria for the EALs. After checking the standard deviation of the normalized gradient method value between the centriod on each clusters ($S_{Ci'} > S_{Cj'}$), we found that the NBL is the cluster of $Ci'$. The lower and top limiter are shown in the Figure R1(b).

[Figure]

**Figure R1-1. The two weighted k-means clustering on 18 Dec 2016 between 2:00-3:00 LST.**

**(a) The first weighted k-means clustering. The results are shown by red and blue hollow and solid points, and their the centroids are represented by larger points of the same colour. The red dash line is shown the height of the limitation of noise. (b) The second weighted k-means clustering. The $S_{Ci'}$ and $S_{Cj'}$ represents the standard deviation of the normalized gradient values in each cluster. The two dash red line is the height restriction of the cluster.**

The word "starfield" appears many times in the article. Do you mean "stratified"? Please confirm it.

**Response:**

Thank you for your suggestion. Yes, the word has been changed as stratified.( Line 151 & 170& 274).

5. From figure 7-2 (c), the NBLH between 21:00 to 22:00 LST is about 640 m (hcjcenter). However, from Figure 7-1, the NBLH of that time period is much higher, why?

**Response:**

In our algorithm, we have defined two constraints to identifying noise as in figure 3 shown. The first is that the noise signal distribution is not clearly stratified ($D_{intra} < D_{sig}$ ), and the second is that the noise is located at a higher height( $h_{ci} > h_{cj}$) and the average standard deviation of the points in noise cluster is smaller than the NBL( $S_{ci} < S_{cj}$). At 21:00-22:00 on April 6, 2017, the distribution of the upper cluster is not meet the requirement of the standard deviation ( $S_{ci}$=0.016, $S_{cj}$=0.033). Therefore, the NBL are in the cluster of the upper layer (cluster in blue).

The content has been changed in P13 Line 274-277.

6. In your CA-GM algorithm, the cluster number is set as two in prior, that is, except NBL, assume that there is only one layer exist above NBL. So what if two or more layers (EALs, cloud layer, or noise layer) exist above NBLH? Besides, I'm concerned that if there is no EALs, cloud layer, or noise layer, does the cluster method affect the NBLH retrieval? Are the NBLHs from the CA-GM similar to that from the GM?

**Response:**

As for the effect two or three more layers, the following results are testing with the real signal.

1) As the example of the 00:00-01:00 Jan 6, 2017. This is a typical multiple layer structure of noise, cloud and NBL.

[Figure]

**Figure R1-2.The example of the multi-layer structure of noise, cloud and NBL.**

**(a) the first weighted k-means clustering (b) the second weighted k-means clustering.**

As a result of the first k-means clustering, seldom noise is located above the cloud layer (Figure R1-2(a)). We set the cluster
as two in prior, there are two groups which indicated the cloud and the possible NBL. According to the criteria to distinguish the cloud layer, the CL and the noise are removed in the upper cluster. Then, we use the second weighted k-means clustering to further identify the NBL. Due to the standard deviation of the GM value in the upper layer is bigger than the lower cluster $(S_{Ci'} > S_{Cj'})$, the final NBL location is at the cluster whose centroid is 598 m.

2) As the example of the 19:00 Dec 19, 2016. There is a scenario of the structure of noise, EALs and NBL.

[Figure]

**Figure R1-3.The example of the structure of noise, EALs and NBL.**

**(a) the first weighted k-means clustering. (b) the second weighted k-means clustering.**

**(c) the backscatter coefficient of the lidar. The distance of two layers are larger than D threshold=100 m.**

As a result of the first k-means clustering, the noise is located at the cluster in red (Figure R1-3(a)). We set the cluster as two in prior, there are two groups which indicated the noise and the possible NBL. According to the criteria to distinguish the noise, noise is removed in the upper cluster. Then, we use the second weighted k-means clustering to further identify the NBL. Due to the backscatter coefficient is excess $1.786 \times 10-3km^{-1}sr^{-1}$. The cluster in red (Figure R1-3(b)) are to be defined as the EALs. Therefore, the NBL is located at the lower cluster (in blue).

**3)** As for the situation of the structure of noise, CLs, EALs and NBL.

There is no real case can be present in this experiment. It is relatively rare in the experiment of this scene. According to the performance of the algorithm, the three possible minima will be located as the CL, EALs ,and the NBL. Seldom will locate as noise affection. If the noise exists and locate over the cloud, the structure can be solved as Figure R1-2. The cloud with noise will be removed by the upper limiter. the four-layer structure has been transformed into a three-layer structure as the Figure R1-3.

As for the complex condition for four or more layers exist, it may cause a certain degree of misjudgement. The algorithm cloud be further developed in seeking for optimal k of the k-means clustering to suitable for more complex condition accurately.

Thank you for your idea.

4) As the situation in the clear condition, on 29 Dec 2016.

If there are no that mentioned three layers , in order to use the CA-GM, it still needs to find the minima in each profile. One point will be layered in obvious NBL, the other two will be removed as the noise cluster. As for the dataset of an hour k-means cluster, the extreme value will be move from the GM. As the shown in Figure R1-4 (a).

[Figure]

**Figure R1-4.The example of the no clear structure of the lidar signal. (a). the time-height cross section of the range-corrected signal (RCS) with four NBLH retrieved method on 29 Dec, 2016. (b-c)The first and second weighted k-means clustering in the 20:00-21:00 LST. (d-e) The first and second weighted k-means clustering in the 21:00-22:00 LST.**

7. Lines 40-53: the authors have clarified some BLH retrieval methods; however, more previous evaluation works should be cited here, for example: McGrath-Spangler et al., 2012, Li et al., 2017.

[1] McGrath-Spangler, E. L., and A. S. Denning (2012), Estimates of North American summertime planetary boundary layer depths derived from space-borne lidar, J. Geophys. Res., 117, D15101, doi: 10.1029/2012JD017615.

[2] H Li, Y Yang*, X-M Hu, Z Huang, G Wang, B Zhang, T Zhang (2017), Evaluation of retrieval methods of daytime convective boundary layer height based on Lidar data, J. Geophy. Res. Atmos., 122, doi: 10.1002/2016JD025620.

Meanwhile, there are some studies have worked to detect cloud or aerosol layers based on lidar data, like Winker et al., 1994, Wang et al., 2001, Li et al., 2017, Dang et al., 2019, should also be cited here, and explain why your work is needed compared to the others'.

[3] Winker, D.M.; Vaughan, M.A. Vertical distribution of clouds over Hampton, Virginia, observed by lidar under the ECLIPS and FIRE ETO programs. Atmos. Res., 1994, 34, 117–133.

[4] Wang, Z.; Sassen, K. Cloud type and macrophysical property retrieval using multiple remote sensors. J. of Appl. Meteorol., 2001, 40, 1665–1683.

[5] H Li, Y Yang*, X-M Hu, Z Huang, G Wang, B Zhang. Application of Convective Condensation Level Limiter in Convective Boundary Layer Height Retrieval Based on Lidar Data. Atmosphere, 2017, 8, 79, doi: 10.3390/atmos8040079.

[6] Dang, R., Yang, Y., Li, H., Hu, X.-M., Wang, Z., Huang, Z., Zhou, T. and Zhang, T.: Atmosphere Boundary Layer Height (ABLH) Determination under Multiple-Layer Conditions Using Micro-Pulse Lidar, Remote Sensing, 11(3), 263, doi:10.3390/rs11030263, 2019.

**Response:**

Thanks for your suggestion. The following reference has been added.

The mentioned reference [1] and [2] had been add in P2.lines 31-32.

'*Multiple approaches have been developed to determine the ABLH based on various observations, including radiosounding, remote sensing, and parameterisation from laboratory experiments (Li et al., 2017b; McGrath-Spangler and Denning, 2012; Nakoudi et al., 2019; Su et al., 2020a).*'

The mentioned reference [3-6] has been add in P2.lines 55-63.

'*The retrieval of BLHs under cloudy conditions is quite challenging. Some researchers have used the threshold of the attenuated scattering ratio (Campbell et al., 2008; Winker and Vaughan, 1994), the ratio of peaks to the base of the range-*

*corrected signal (RCS) (Wang and Sassen, 2001) to locate cloud tops and bases, while others have employed the objective upper limit of the convective condensation level (CCL)(Li et al., 2017a), as well as the analysis of signal continuity and the classification of whether the cloud caps the ABLH or is decoupled from the ABL (Dang et al., 2019b). The height restriction has significant advantages in removing the influence of clouds. Elevated aerosol layers (EALs) are characteristically similar to the aerosol trapped in ABL, using the threshold of lidar backscatter coefficient can distinguish them (Dubovik et al., 2002;*

*Hänel et al., 2012; Peng et al., 2017). More instrument and multi-wavelength lidar systems are combined to obtain more accurate results to identified the EALs (Liu et al., 2019; Ortega et al., 2016).'*

**Minor revision:**

1.Line 23: Change "continues" to "continuous".

2.Line 30: Change "on observation" to "based on various observations".

**Response:**

The word had been changed in P1.lines 23.

The word had been changed in P2.lines 30.

3.Line 34: coefficient between what?

**Response:**

The correlation coefficient between lidar retrieval algorithm and radiosonde is lower under stable conditions due to a complex aerosol structure that increases the difficulty of NBLH retrieval.

The sentence have rephrased P2.lines 33-35.

*'The stable condition shows further agreement between lidar and radiosonde than the unstable condition because of the complex aerosol structure that complicates NBLH retrieval (Emeis and Schäfer, 2006; Martucci et al., 2007; Sawyer and Li, 2013). '*

4.Lines 51-52: The sentence is difficult to understand.

**Response:**

The sentence have been revised in P2.lines 49-54.

*'Some graph theory methods, such as the extended Kalman filter (Banks et al., 2014), Pathfinder and PathfinderTURB (de Bruine et al., 2017; Poltera et al., 2017), k-means clustering (Liu et al., 2018; Toledo et al., 2014), and The STRAT-2D algorithm (Haeffelin et al., 2012) have been proposed to yield promising results via an automated method that reduces the*

*incorrect detection of ABLH. However, these techniques strongly depend on the vertical distribution of particle layers (aerosols and clouds) and are unsuitable for use under complicated multilayer conditions (Granados-Muñoz et al., 2012). '*

5.Line 55: The fluctuation of NBLH, such statement is not completed. Line 63: Delete "in the experiment".

**Response:**

Thank you for your suggestion.

The word have been changed in P2.lines 65.

'Digressing from these previous efforts to estimate the ABLH, we herein present a new approach—cluster analysis of the gradient method (CA-GM)—to overcome the multilayer structure and remove the noise fluctuation of NBLH with raw data resolution.'

And the word has been delete.

**6.Line 87: the value of turbulence? Such statement is incorrect.**

**Response:**

Thank you for your suggestion. The word has been changed as turbulence intensity in P4 Line104-105.

[revised manuscript text omitted]

---

## Author Comment (AC4) · 15 Sep 2020

**Responds to Anonymous Referee #3:**

**General comments:**

The manuscript "A novel Mie lidar gradient cluster analysis method of nocturnal boundary layer detection during air
5   pollution episodes" presented a new algorithm to retrieve the nocturnal boundary layer height (NBLH), based on cluster
analysis of gradient method, using 39 days lidar observations. The radiosonde data were used to evaluate its performance of
NBLH retrieval, and results show that the presented algorithm had a better agreement than the other 3 methods (GM, WCT,
CRGM). A comparison of this new methods with the other 3 methods were also analyzed and discussed, using a 256 hours
data set. The presented method is promising for improving the NBLH retrieval, and results look interesting. However, I don't
10   think the current form can clearly deliver the information, and a number of point must be clarified. Major changes are
needed, and the writing of the paper must be improved, before the manuscript can be considered for publication. Please see
my comments below.

**Response:**

15   Thanks a lot for your reviews on our manuscript entitled "A novel Mie lidar gradient cluster analysis method of nocturnal
boundary layer detection during air pollution episodes (ID: amt-2020-167). We have revised the manuscript according to the
comments, the language has been polished by Elsevier Language Editing Services. Moreover, the comprehensive reference
and the discussion of the results, and the limitation and the uncertainty of the algorithm have been added. The details are
shown as follows.

20

**Specific comments:**

1.The presented method can only be applied for the BIT-lidar or can be used for other elastic lidars? Why you used "Mie
lidar" in the title? I think this method is not only valid for Mie lidar. The conditions/constraints of using such method should
be discussed.

25   **Response:**

1) In my opinion, this method can be used in other elastic lidar systems. The implement of the algorithm is needed a dataset
of the RCS gradient with height more than 30 min collection within an hour. I think it can be used for other elastic lidar.
Therefore, we have changed the title as a n*ovel lidar gradient cluster analysis method of nocturnal boundary layer detection
during air pollution episodes*.

30   2) The reason why I used "Mie lidar" is that our lidar system (BIT-lidar) system is the Rotation Raman Mie system, and I
use the Mie channel signal to detect the atmospheric boundary layer.

3) The condition/constraints are added to the discussion.

'*The limitation of the CA-GM is based on the nocturnal boundary layer is stable, hence, we can calculate the distribution of
the minima gradients of the RCS in an hour interval to use weighted k-means clustering to work as height restriction to the*

35 *layers. Secondly, based on the limitation of the lidar system. The lower limit of the BIT-lidar is around 300 m. Too shallow of*
*nocturnal boundary layer height (NBLH) are not be detectable. Thirdly, the method should be used in the high SNR*
*condition, such as night-time and air pollution.'*

In this study, the raw data resolution is used to get a high time & vertical resolution, any comparison with other methods?
40 Have you used any vertical smoothing? What is the final time & vertical resolution? If reader want to apply this method to
another lidar system, what's the limitation? Some discussions are need.

**Responds:**

1)The raw data time resolution of lidar system is 50 s and the vertical resolution is 2.5 m. It is a relative high time and
vertical resolution. Other research using 2 min average signal (Su and Patrick McCormick, 2019),15 min time-averaged
45 signal (Martucci et al., 2010) or even 30 min time-average(Tsaknakis et al., 2011) for elastic lidar system tracing the aerosol
distribution.

2) In our algorithm, we use the Savitzky-Golay smooth method at the preprocess of RCS. The final time & vertical is the raw
resolution, but the effective time resolution is 30 min. In order to implement the algorithm, we should collect at least 30 min
of the dataset for the RCS gradient with height within an hour. Therefore, we have enough dataset for cluster analysis
50 (CA).The CA determines the NBL by taking into account the overall set of observations of a given profile, which can be
considering as the effective time resolution.

3) The limitation of the implement of the method are added at the discussions part. P17 332-335.

2. No uncertainty/error study is presented. Such information should be added.
55 **Responds:**

Thank you for your suggestion. The testing with the real signal are shown below, and can be added at the Supplement.

1)The testing with the real signal are shown below.

Use the RCS(z) signal, and randomly noised $RCS^{noised}(z)$ by the expression:

$$RCS^{noised}(z) = RCS(z) + [\alpha \times \chi(z)] \quad (R3\text{-}1)$$

60 Where $\chi(z)$ is the random noise function taking values between 0 and 1, z is the height, and $\alpha$ is a varying parameter as
introduced in Eq (R3-1) to produces different levels of noise.

[Figure]

**Figure R3-1. The real lidar RCS for the heavily polluted case (17 Dec 2016 20:00-21:00 LST ).**

**(a-c) three noise level cases, (d-f) with the gradient of RCS, and (g-i) the first weighted k-means clustering.**

[Figure]

**Figure R3-2 RMSE between the WCT and the other three algorithms (GM,CRGM and CA-GM )**

As a result of the figure R3-2 shows, the CA-GM has less RMSE than GM at the ratio of 1%-4%. The figure R3-1 (g-h) shows similar groups in different range of noise affection. However, the clustering changes at the results of R3-1(i). Due to the noise distribution of the signal, the centriod of the cluster will get higher and lose the ability to restrict the changes of GM. The difference in the NBL top is found with the noise level 4% case, they are lower than 1% in respect to the estimate for the case with raw signal.

[Figure]

**Figure R3-3. The real lidar RCS for the cloud case (5 Jan 2016 00:00-1:00 LST).**

**(a-d) the different ratio of strength of the cloud layer intensity. (e-h) the first weighted-means clustering**

Add the signal of the cloud layer on the raw data, the ratio of the intensity for cloud layer changes from -40% to 40%. As the figure shown the first k-means clustering in figure R3-3(e-h),the intensity of the cloud layer will not influence the CA-GM.

In summary, these results indicate that the degree of estimation of the NBL top by applying CA is weakly affected by the signal noise. In fact,  a few NBLH depending on the value of the RCS gradient in a discrete point. CA determines the NBL by taking into account the overall set of observations of a given point, thus decreases the dependence of the method on the RCS values in single moment. The intensity of the CLs changes ±40% and will not affect the cluster of the CA-GM, it can be significant stratified due to the relative significantly signal difference on the backscatter signal. As for the EALs, the strict threshold will defined the EALs accurately. Therefore, the CA-GM approach is able to accurately obtain the NBLH with the effect of noise, cloud layers and elevated aerosol layers.

We add the uncertainty analysis in the discussion in P17 Line 336-341.

3. The description of methodology is not clear, please revise it.

90 L88, explain more about the assumption.

**Responds:**

The content has been added in P4 Line 102-105.

  *'The NBL shows more complex internal structure at night, the particulate can be used as an important indicator of atmospheric layering because its vertical distribution is strongly affected by the thermal structure of the atmosphere (Neff*
95 *and Coulter, 1986).The assumption of the NBL at which the aerosol concentration and turbulence intensity are significantly higher in the NBL than in the free atmosphere (FA)(Dang et al., 2019; Wang et al., 2020).'*

**Fig2, add legend for red line, grey lines, colour circles etc. are GM peaks from the red line? More description needed.**

Responds:

100 Thank you for your suggestion.

These contents have been added to the article in P6.line 138-142.

[Figure]

  *'Figure 2. The theoretical schematic of the weighted-k means clustering. (a)The real profile of a lidar RCS( light gray line) and the hour averaged RCS (black line). (b) The gradient of RCS (light gray line), the hour averaged gradient RCS (black line), and the three*
105 *minima in the profile (yellow points). (c) The distribution of the gradient minima within an hour. (d-e) The results obtained by standard k-means and weighted k-means clustering, where two clusters are differentiated, as shown by red and blue hollow and solid points, respectively.'*

L122, "three minima peaks", and L145 "three gradient minima", do they refer to the same information? Please clarify which
110 minima criterion you used, the 3 minima gradient values of RCS? Or the peaks with minima values? You can also add these minima by the markers in figure 2.

**Responds:**

The content has been add in P6 Line 144-146

Yes, the three minima peaks and the three gradient minima are both the same information. The method of finding the three

115 minima of RCS gradient is collecting the minimum points by a window of 25 m, and sort the top three minimum points.

L144, describe more about the reference height.

**Responds:**

We are choosing as in the Fernald algorithm by choosing a molecular reference so that $\beta_m{}^{aero} \ll \beta_m{}^{mol}$.

120 The reference has been added in P7 Line 167.

Fig4, you can put all other methods using different colour/marker.

**Responds:**

The Figure4 has been changed in P9 Line 200.

[Figure]

125

*'Figure 4. Comparison between the radiosonde-determined and lidar-retrieved measurement of NBLH in gradient method (GM) (red circle), Wavelet covariance transform transition method (WCT)(blue triangle), cubic root gradient method (CRGM) (orange star) and cluster analysis of gradient method (CA-GM) (black circle). The correlation coefficients is represented by R. The black solid line is the 1:1 line.'*

130

L225, Are you sure it is a cloud layer? RCS looks very weak for this layer. It could be a lofted aerosol layer. If it is not a cloud layer, another case should be presented in this section.

**Responds:**

Thank you for your suggestion. It is really hard to use single wavelength lidar to layer the classification of the aerosol and

135 cloud layers. According to the (Zhao et al., 2014), The maximum and the minima of the F(z) are donated as T and D, respectively. When z is below 3 km, layers are classified as clouds when T>3 or D<-7; As the following figure shows, the T just overpass the threshold. Therefore, it cloud be a weak cloud layer. If you think it is still not reasonable, I can change for another example.

[Figure]

140 **Figure R3-3. The gradient of logarithmic ranged correction signal**

**Technical corrections:**

L12, 39 days is not a "long-term", maybe another expression.

**Responds:**

The long-term have been changed as 39-d in P1 Line12.

145

L32-33, rephrase the sentence.

**Responds:**

The sentence have change in P9 Line 49-54.

*'Some graph theory methods like extend Kalman filter (Banks et al., 2014), Pathfinder and PathfinderTURB (de Bruine et al.,*
150 *2017; Poltera et al., 2017), and k-means clusters (Liu et al., 2018; Toledo et al., 2014) are proposed to promising results using an automated method which reduces incorrect detection of the ABLH. However, all these methods will higher cause uncertainty in ABLH identification when encountering a multiple layer vertical structure. However, those techniques depend strongly on the vertical distribution of particle layers (aerosols and clouds) and are not suitable for dealing with complicated multiple-layer conditions* (Granados-Muñoz et al., 2012).'

155

L67, provide the vertical resolution of radiosonde.

**Responds:**

The L-band radiosonde provided fine-resolution profile of temperature, pressure, relative humidity, wind speed and direction twice a day at 08:00 and 20:00 local standard time (LST) (Guo et al., 2016).The sample time resolution is 1.2 seconds. The
160 vertical resolution varies from site to site.

Th content has been added in P3 Line 79.

*'The vertical temporal resolution was 1.2 s, and the vertical resolution is less than 20 m.'*

L73, add "gradient" for PTG

165 **Responds:**

Thank you for your suggestion, the word has been add at P2.Line 86.

L76, is BIT-lidar rotational Raman–Mie lidar, but in this study you only use the elastic channel?

**Responds:**

170 Yes. We use the Mie signal only.

L79, after the overlap correction, what's the lower limit for BIT-lidar?

**Responds:**

The lower limit of BIT-lidar is 300 m. The content has been added at discussion.

175

L89, NBL top. Add "top" here.

**Responds:**

The word has been added in P5. Line 105.

180 L115, change hw to hnor

**Responds:**

The words has been changed in P5. Line 132.

L127, "the noise . . . be affected" do you mean the accuracy can be affected. L130, add "layers" for EALs.

185 **Responds:**

Yes, the sentence has been changed in P6.Line 150. And the word has been added at P6. Line 60.

L170, what do you means here "with all algorithm"? L206, any value for this "low SNR condition"?

**Responds:**

190 With the other three algorithm (GM,WCT and CRGM).The sentence has been changed in P8 Line 193-194.

The Brooks not mentioned the specific value for the low SNR condition for WCT method(Brooks, 2003).

L208, please specify which "improvement".

**Responds:**

195    The improvement has been express in P11 Line 233-235.

L282, "was automatics developed"?

**Responds:**

Thank you for your suggestion. I delete the words.

200

L283, "high time resolution", please specify it. is it equal to the lidar vertical resolution?

**Responds:**

The inaccurate expression has been changed as the lidar raw resolution.P17 Line 312.

205    Thank you so much for your reviewing! We deeply appreciate your recognition of our research work.

**Reference**

Banks, R. F., Tiana-Alsina, J., María Baldasano, J. and Rocadenbosch, F.: Retrieval of boundary layer height from lidar
210    using extended Kalman filter approach, classic methods, and backtrajectory cluster analysis, edited by A. Comerón, E. I.
Kassianov, K. Schäfer, R. H. Picard, K. Stein, and J. D. Gonglewski, p. 92420F, Amsterdam, Netherlands., 2014.

Brooks, I. M.: Finding boundary layer top: Application of a wavelet covariance transform to lidar backscatter profiles,
Journal of Atmospheric and Oceanic Technology, 20(8), 1092–1105, 2003.

de Bruine, M., Apituley, A., Donovan, D. P., Klein Baltink, H. and de Haij, M. J.: Pathfinder: applying graph theory to
215    consistent tracking of daytime mixed layer height with backscatter lidar, Atmospheric Measurement Techniques, 10(5),
1893–1909, doi:10.5194/amt-10-1893-2017, 2017.

Dang, R., Yang, Y., Hu, X.-M., Wang, Z. and Zhang, S.: A Review of Techniques for Diagnosing the Atmospheric
Boundary Layer Height (ABLH) Using Aerosol Lidar Data, Remote Sensing, 11(13), 1590, doi:10.3390/rs11131590, 2019.

Granados-Muñoz, M. J., Navas-Guzmán, F., Bravo-Aranda, J. A., Guerrero-Rascado, J. L., Lyamani, H., Fernández-Gálvez,
220    J. and Alados-Arboledas, L.: Automatic determination of the planetary boundary layer height using lidar: One-year analysis
over southeastern Spain: DETERMINATION OF THE PBL HEIGHT, Journal of Geophysical Research: Atmospheres,
117(D18), n/a-n/a, doi:10.1029/2012JD017524, 2012.

Guo, J., Miao, Y., Zhang, Y., Liu, H., Li, Z., Zhang, W., He, J., Lou, M., Yan, Y., Bian, L. and Zhai, P.: The climatology of
planetary boundary layer height in China derived fromradiosonde and reanalysis data, Atmos. Chem. Phys., 16(20), 13309–
225    13319, doi:10.5194/acp-16-13309-2016, 2016.

Liu, B., Ma, Y., Liu, J., Gong, W., Wang, W. and Zhang, M.: Graphics algorithm for deriving atmospheric boundary layer
heights from CALIPSO data, Atmospheric Measurement Techniques, 11(9), 5075–5085, doi:10.5194/amt-11-5075-2018,
2018.

Poltera, Y., Martucci, G., Collaud Coen, M., Hervo, M., Emmenegger, L., Henne, S., Brunner, D. and Haefele, A.: PathfinderTURB: an automatic boundary layer algorithm. Development, validation and application to study the impact on in situ measurements at the Jungfraujoch, Atmospheric Chemistry and Physics, 17(16), 10051–10070, doi:10.5194/acp-17-10051-2017, 2017.

Su, J. and Patrick McCormick, M.: Using multi-wavelength Mie–Raman lidar to measure low-level cloud properties, Journal of Quantitative Spectroscopy and Radiative Transfer, 237, 106610, doi:10.1016/j.jqsrt.2019.106610, 2019.

Toledo, D., Córdoba-Jabonero, C. and Gil-Ojeda, M.: Cluster Analysis: A New Approach Applied to Lidar Measurements for Atmospheric Boundary Layer Height Estimation, Journal of Atmospheric and Oceanic Technology, 31(2), 422–436, doi:10.1175/JTECH-D-12-00253.1, 2014.

Tsaknakis, G., Papayannis, A., Kokkalis, P., Amiridis, V., Kambezidis, H. D., Mamouri, R. E., Georgoussis, G. and Avdikos, G.: Inter-comparison of lidar and ceilometer retrievals for aerosol and Planetary Boundary Layer profiling over Athens, Greece, Atmospheric Measurement Techniques, 4(6), 1261–1273, doi:10.5194/amt-4-1261-2011, 2011.

Wang, H., Li, Z., Lv, Y., Zhang, Y., Xu, H., Guo, J. and Goloub, P.: Determination and climatology of the diurnal cycle of the atmospheric mixing layer height over Beijing 2013–2018: lidar measurements and implications for air pollution, Atmospheric Chemistry and Physics, 20(14), 8839–8854, doi:10.5194/acp-20-8839-2020, 2020.

Zhao, C., Wang, Y., Wang, Q., Li, Z., Wang, Z. and Liu, D.: A new cloud and aerosol layer detection method based on micropulse lidar measurements: MPL based Aerosol and Cloud Detection, Journal of Geophysical Research: Atmospheres, 119(11), 6788–6802, doi:10.1002/2014JD021760, 2014.